



**1** **A SOUTH ATLANTIC ISLAND RECORD UNCOVERS SHIFTS IN**

**2** **WESTERLIES AND HYDROCLIMATE DURING THE LAST GLACIAL**

**4** **Svante Björck**[1,2]**, Jesper Sjolte**[1]**, Karl Ljung**[1]**, Florian Adolphi**[1,3]**, Roger Flower**[4]**, Rienk**

**5** **H. Smittenberg**[2]**, Malin E. Kylander**[2]**, Thomas F. Stocker**[3]**, Sofia Holmgren**[1]**, Hui Jiang**[5]**,**

**6** **Raimund Muscheler**[1]**, Yamoah K. K. Afrifa**[6]**, Jayne E. Rattray**[7]**, Nathalie Van der**

**7** **Putten**[8]

**9** [1]Department of Geology, Lund University, SE-22362 Lund, Sweden

**10** [2]Department of Geological Sciences and the Bolin Centre for Climate Research, Stockholm

**11** University, SE-10691 Stockholm, Sweden

**12** [3]University of Bern, Physics Institute, Climate and Environmental Physics, Sidlerstrasse 5, CH-3012

**13** Bern, Switzerland

**14** [4]Department of Geography, University College London, London WC1E 6BT, UK

**15** [5]Key Laboratory of Geographic Information Science, East China Normal University, 200062

**16** Shanghai, PR China

**17** [6]School of Geography, Earth and Environmental Sciences, University of Birmingham, Edgbaston, B15

**18** 2TT, UK

**19** [7]Department of Biological Sciences, University of Calgary, Calgary, Canada

**20** [8]Earth and Climate Cluster, Faculty of Science, Vrije Universiteit, Amsterdam, The Netherlands

**22** **Correspondence:** Svante Björck (svante.bjorck@geol.lu.se)

**23** Abstract

**24** The period 36-18 ka was a dynamic phase of the last glacial, with large climate shifts in both

**25** hemispheres. Through the bipolar seesaw, the Antarctic Isotope Maxima and Greenland DO

**26** events were part of a global "concert" of large scale climate changes. The interaction between

**27** atmospheric processes and Atlantic meridional overturning circulation (AMOC) is crucial for

**28** such shifts, controlling upwelling- and carbon cycle dynamics, and generating climate tipping

**29** points. Here we report the first temperature and humidity record for the glacial period from

**30** the central South Atlantic (SA). The presented data resolves ambiguities about atmospheric

**31** circulation shifts during bipolar climate events recorded in polar ice cores. A unique lake

**32** sediment sequence from Nightingale Island at 37°S in the SA, covering 36.4-18.6 ka, exhibits

**33** continuous impact of the Southern Hemisphere Westerlies (SHW), recording shifts in their



position and strength. The SHW displayed high latitudinal and strength-wise variability 36-31
ka locked to the bipolar seesaw, followed by 4 ka of slightly falling temperatures, decreasing
humidity and fairly southern westerlies. After 27.5 ka temperatures decreased 3-4°C, marking
the largest hydroclimate change with drier conditions and a variable SHW position. We note
that periods with more intense and southerly positioned SHW are correlated with periods of
increased $CO_2$ outgassing from the ocean. Changes in the cross-equatorial gradient during
large northern temperature changes appear as the driving mechanism for the SHW shifts.
Together with coeval shifts of the South Pacific westerlies, it shows that most of the Southern
Hemisphere experienced simultaneous atmospheric circulation changes during the latter part
of the last glacial.

## 1 Introduction

The Southern Hemisphere Westerlies (SHW) is a major determinant of hydroclimate in the
Southern Hemisphere (SH). In coupling marine and atmospheric processes, they are thought
to have played a pivotal and multi-faceted role during and at the end of the last ice age by
triggering changes in ocean-atmosphere $CO_2$ fluxes by physical processes (Saunders et al.,
2018; Toggweiler and Lea, 2010) and Fe fertilization of the Southern Ocean through varying
dust deposition (Lamy et al., 2014; Martin and Fitzwater, 1988; Martínez-García et al., 2014),
as well as regulating the salt and heat leakage from the Agulhas current to the Atlantic
meridional overturning circulation (AMOC) (Bard and Rickaby, 2009).  In addition, changes
in AMOC, SHW strength and position, and Southern Ocean upwelling seem to have been
important mechanisms for different glacial $CO_2$ modes (Ahn and Brook, 2014). The position
of the SHW during glacial times is debated with some arguing for a northward displacement
(Toggweiler et al., 2006), while others argue for a southward move (Sime et al., 2013, 2016)
during the Last Glacial Maximum (LGM), relative to the present. Holocene data also suggest
an expanding-contracting SHW zone (Lamy et al., 2010). With these multiple scenarios the





pattern of SHW shifts and their detailed role for ocean ventilation and the global carbon cycle
remains unclear. It is postulated that the SHW moved in concert with rapid climate shifts
recorded in Greenland ice cores known as Dansgaard-Oeschger (DO) cycles (Markle et al.,
2016), and that these shifts are part of inter-hemispheric climate swings involving heat
exchange between the hemispheres through the atmosphere and the ocean, with atmospheric
heat fluxes partly compensating anomalous marine heat fluxes (Pedro et al., 2016). Whether
SHW zonal shifts only occurred in the Pacific sector of the Southern Ocean (Chiang et al.,
2014) or if they occurred throughout the SH is another crucial question (Ceppi et al., 2013).
Other key climate issues relate to the effects and areal extent of the bipolar seesaw mechanism
(Broecker, 1998; Stocker and Johnsen, 2003) and any signs of an early and long temperature
minimum at southern mid-latitudes matching Antarctic LGM (EPICA Community Members
et al., 2006). The lack of climate proxy records directly reflecting atmospheric conditions in
the central South Atlantic means that such information at these latitudes during the glacial are
primarily based on remote proxy records or climate model simulations. This results in a
largely unconstrained understanding of glacial conditions over vast parts of the mid-South
Atlantic, especially between 20-50°S where archives reflecting atmospheric processes are
absent.



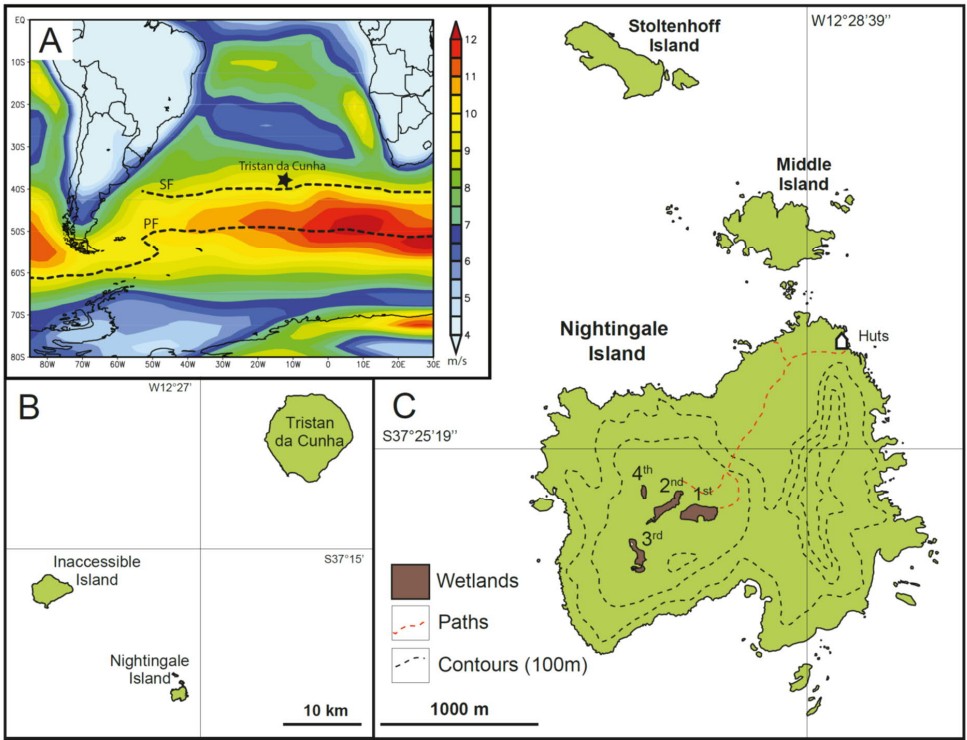

**Figure 1.** (A) The position of the Tristan da Cunha island group in the South Atlantic, the 1000mb mean annual wind speed (m/s) for 1980-2010 according to NCEP/NCAR reanalysis data indicating yellow-red colors for the zone of the Southern Hemisphere Westerlies, and the positions of the Subtropical Front (SF) and the Polar Front (PF) as dashed lines. (B) The three main islands of the Tristan da Cunha island group. (C) The position and size of the four overgrown lake basins, so-called ponds (1P-4P), on Nightingale Island with 100 m contour lines.

## 2 Study site

The Tristan da Cunha island group (TdC) at 37.1° S (Fig. 1) sits strategically at the northern

boundary of the SHW (Fig. 1A), a few degrees north of the Subtropical Front (SF), where sea

surface temperatures (SST) and salinities decrease by 3-4°C and 0.3 per mil, respectively.

Annual mean air temperature and precipitation are 14.3°C and approximately 1500 mm,

respectively, with highest precipitation in austral winter when the SHW impact is largest. The

record presented here is from 1st Pond (1P), an overgrown crater lake (200x70 m, 207 m a.s.l.)



in the central part of Nightingale Island (NI) (Fig. 1C and Fig. 2), a volcanic island dominated
by trachytic bedrock. Its drainage area is about twice the size of today`s peat-bog and is thus
sensitive to changes in the precipitation/evaporation balance (P/E). Previous studies from NI
show that the area experienced shifts in precipitation during the Holocene (Ljung and Björck,
2007) and partly also during the Last Termination (Ljung et al., 2015), mainly attributed to the
changing impact of the SHW. These data also indicate a southerly displacement of the
Intertropical Convergence Zone (ITCZ) during the Heinrich 1 event (H1), and warming in the
South Atlantic as a consequence of reduced AMOC, causing the lake basin to dry out, creating
a hiatus between 18.6-16.2 ka (Ljung et al., 2015). Here we present a multi-proxy study of the
sediments that accumulated before this hiatus dating to 36.4-18.6 ka, covering the younger
part of Marine Isotope Stage 3 (MIS 3) and most of MIS 2, a climatically very dynamic
period with Antarctic Isotope Maxima, DO and H events. In spite of its fairly northern
position in relation to Antarctica we hypothesize that TdC was impacted by such events in
terms of shifts of SHW, which we aim to test by using a suite of proxies.



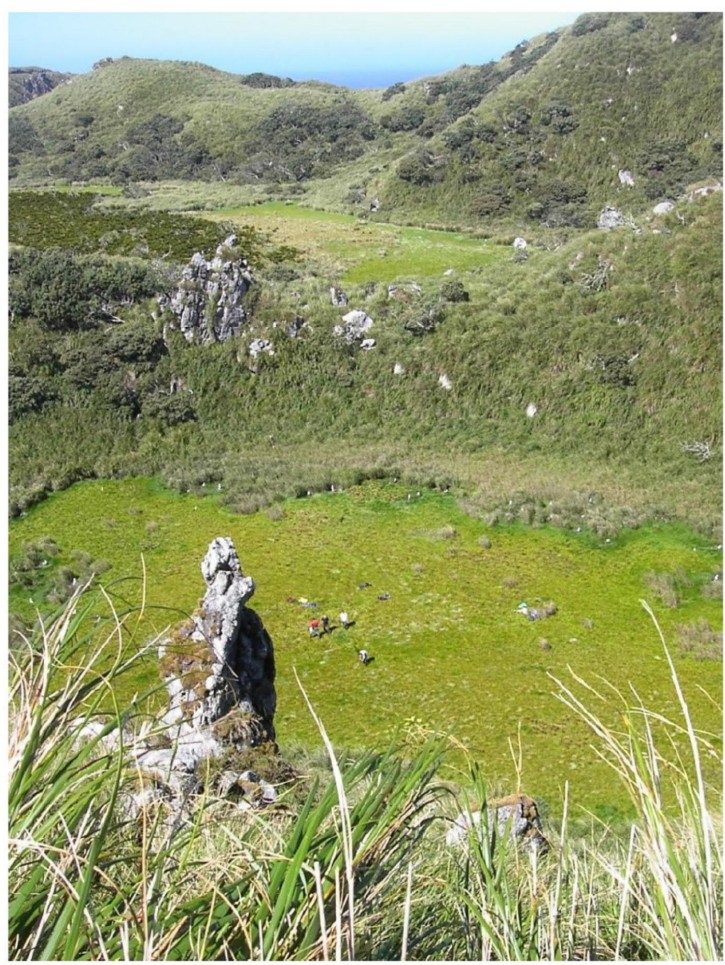


**Figure 2.** Photograph from Nightingale Island. The over-grown lake basins of 1$^{st}$ and 2$^{nd}$ Pond are shown, with the higher situated 1$^{st}$ Pond in the background, seen towards southeast. Note the albatross chicks (white dots) and the four persons on 2$^{nd}$ Pond as scale. Photo S. Björck.

## 3 Material and methods

### 3. 1 Field work, handling of cores and sample collection

Two weeks of field work on NI were carried out in February 2010 and drilling was carried out

using Russian chamber samplers providing 1 m long cores (Ø=50 and 75 mm) with overlaps

of 15-50 cm between each cored section. The ketch *Ocean Tramp* provided the transport from

the Falkland Islands to TdC and back to Uruguay. In order to penetrate as deep as possible



into the very stiff sediments a chain-hoist was used for coring the deeper parts of the
sequences. The sediments were described immediately in the field before being wrapped in
plastic film and PVC tubes. Upon arrival in Uruguay the cores were transported to the
Geology Department in Lund where they were stored in a cold room. Before sub-sampling for
the different proxy analyzes, the field-based lithostratigraphy and correlations between
individual core sections were adjusted in the laboratory. This was aided by magnetic
susceptibility ($\kappa$) measurements, which give a relative estimate of the magnetic mineral
concentration, to confirm and adjust the visual correlation between overlapping core
segments.

## 3.2 Radiocarbon dating and age model

The radiocarbon dated material consisted of 1 cm thick, organic-rich, bulk sediment. All 41
dated samples were pre-treated and measured at the Lund University Radiocarbon Dating
Laboratory with Single Stage Accelerator Mass Spectrometry (SSAMS). The age model was
constructed using the OxCal software package (Bronk Ramsey, 1995, 2009a). To minimize
subjective user input we ran the age model with a general outlier model (Bronk Ramsey,
2009b), and a variable k-value that lets the model itself determine the sedimentation rate
variability (Fig. 3). For calibration we use the Southern Hemisphere calibration dataset,
SHCal13 (Hogg et al., 2013)).

## 3. 3 Measurements for magnetic susceptibility

Magnetic susceptibility ($\kappa$) was measured using a Bartington MS2E1 high resolution surface
scanning sensor coupled to a TAMISCAN automatic logging conveyor. Measurements were
carried out on non-sampled half cores and with a resolution of 5 mm and with results shown





in 10$^{-6}$ SI units. The magnetic susceptibility gives a relative estimate of the ability of the
material to be magnetized, i.e. the magnetic mineral concentration.

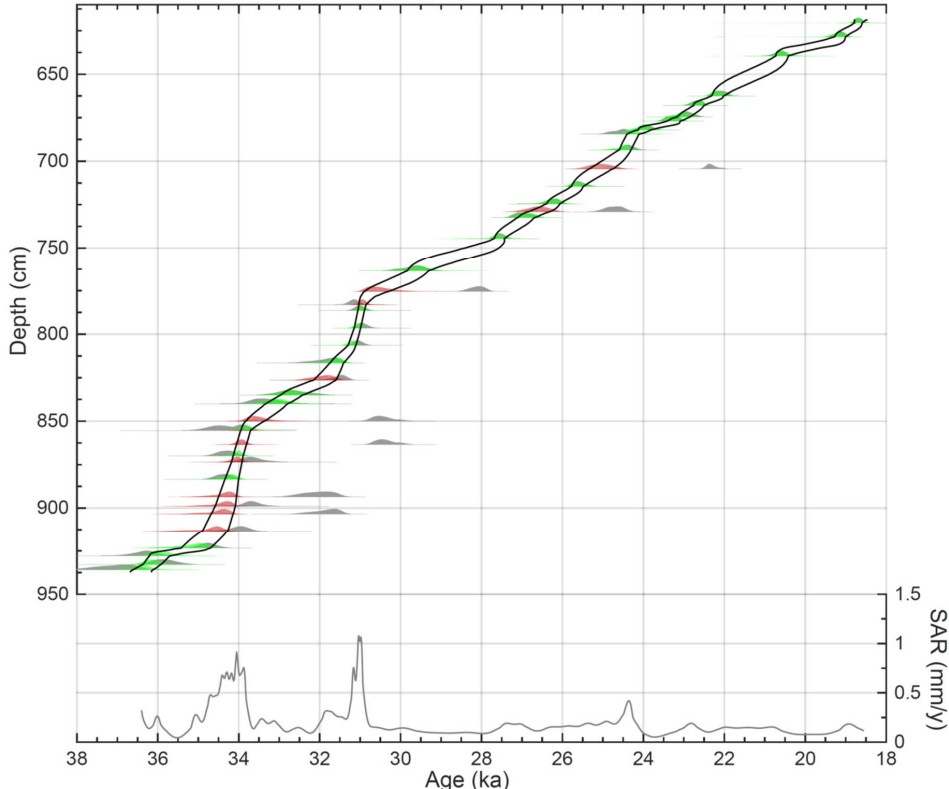


**Figure 3.** Age model for the sediments at 1$^{st}$ Pond, Nightingale Island. Top panel: Radiocarbon based
age-depth model (black lines encompass the 68.2% probability interval). The patches indicate the
calibrated probability distributions of each radiocarbon date for un-modelled (single) dates (grey patch),
and their posterior distributions when modeled as a P-sequence: Green patches indicate agreement
indices of >60% and red patches agreement indices of <60%, i.e., outliers. Bottom panel: Sediment
accumulation rates (mm a$^{-1}$) based on the mean age-depth model shown in the top panel.

### 3. 4 XRF analyses

A handheld Thermo Scientific portable XRF analyzer (h-XRF) Niton XL3t 970 GOLDD+ set
in the Cu/Zn mining calibration mode was used. The instrumentation provides highly accurate
determinations for major elements (Helfert et al., 2011). All analyses were performed on
freeze-dried sediments from the 1P cores using an 8 mm radius spot size in order to obtain



representative values. The elemental detection depends partly on the duration of the analysis
at each point; this is especially true for the lighter elements such as Mg, Al, Si, P, S, Cl, K and
Ca. For this reason the measurement time of each sample was set to 6 minutes. Although a
larger suite of elements was acquired, we have chosen to work with Al, Si, P, S, K, Ca, Ti,
Mn, Fe, Rb, Sr and Zr. These elements were selected based on their analytical quality (i.e.,
level above the detection limit) and with the help of Principal Component Analysis (PCA).
PCA was made using JMP 10.0.0 software in correlation mode using a Varimax rotation.
Before analysis all data were converted to Z-scores calculated as $(X_i-X_{avg})/X_{std}$, where $X_i$ is
the normalized elemental peak areas and $X_{avg}$ and $X_{std}$ are the series average and standard
deviation, respectively, of the variable $X_i$. A Varimax rotation allocates into the components
variables which are highly correlated (sharing a large proportion of their variance) – imposing
some constrains in defining the eigenvectors. By grouping together elements showing similar
variation, the chemical signals tend to be clearer and key elements are better identified. To
simplify the interpretation of our principal components (PC) we employ a modified Chemical
Index of Alteration (CIA), see Fig. 4D, as defined by Nesbitt and Young (1982): CIA =
$[Al_2O_3/(Al_2O_3 + CaO + NaO + K_2O)]$ x 100. This index expresses the relative proportion of
$Al_2O_3$ to the more labile oxides and is an expression of the degradation of feldspars to clay
minerals. Since we have no NaO data we call it a *modified* CIA.
**3.5 C and N analyses**
Dried and homogenized samples every 1-2 cm were analysed with a Costech Instruments ECS
4010 elemental analyzer. The accuracy of the measurements is better than ± 5% of the
reported values based on replicated standard samples. C/N atomic ratios were obtained by
multiplying by 1.167.

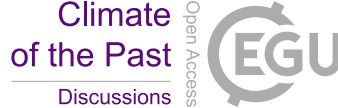

### 3.6 $^{13}$C and $^{15}$N analyses

Dried homogenized bulk samples were measured using a ThermoFisher DeltaV ion ratio mass

spectrometer. The isotopic composition of samples is reported as conventional δ-values in

parts per thousand relative to the Vienna Pee Dee Belemnite ($^{13}$C) and atmospheric N ($^{15}$N):

$\delta_{sample}$ (‰) = [($R_{sample}$-$R_{standard}$)/($R_{standard}$)] x 1000 where R is the abundance ratio of $^{13}$C/$^{12}$C in

the sample or in the standard.

### 3.7 Pollen analyses

Sixty-four levels were sub-sampled and analysed for their pollen content. Pollen samples of 1

cm$^3$ were processed following standard method A as described by Berglund and Ralska-

Jasiewiczowa (1986) with added *Lycopodium* spores for determination of pollen concentration

values. Counting was made under a light microscope at magnifications of x400 and x1000.

The aim was to count at least 500 pollen grains in every sample, which was almost achieved

(mean sum of 565 pollen grains and mean sum of 870 pollen grains and spores). Identification

of pollen grains and spores was facilitated by published photos (Hafsten, 1960), standard

pollen keys (Moore et al., 1991) and a small collection of type slides from Tristan da Cunha

borrowed from The National History Museum in Bergen. The pollen percentage diagram (Fig.

5) was plotted in C2 (Juggins, 2007). Warm/cold pollen ratios were calculated as $W_p/W_p$ +

$C_p$, where warm pollen types ($W_p$) are from plants only found below 500 m a.s.l. and cold

pollen types ($C_p$) are from plants only found above 500 m a.s.l.

### 3.8 Diatom analyses and diatom environmental ratios

levels of 0.5 cm thick sediment segments were sub-sampled to analyse their diatom

content. For preparation of diatom slides ~ 200 mg freeze-dried sediment was oxidized with

15% $H_2O_2$ for 24 hours, then 30% $H_2O_2$ for a minimum of 24 hours, and finally heated at

90°C for several hours. A known quantity of DVB (divinylbenzene) microspheres was added

to 200 μL aliquots of the digested and cleaned slurries in order to estimate diatom

concentrations (Battarbee and Keen, 1982). The diatoms were mounted in Naphrax® medium



(refractive index = 1.65). 300 valves or more per sample were counted in most samples and
identified largely using published diatom floras (Krammer and Lange-Bertalot, 1986; Lange-
Bertalot, 1995; Le Cohu and Maillard, 1983; Moser et al., 1995; Van de Vijver et al., 2002).
Diatom results are expressed as relative % abundance of each taxon (Fig. S3) and also as total
concentrations of valves per g dry sediment.

Freshwater diatom species are excellent indicators of water quality, particularly

of pH, conductivity and dissolved nutrients (Battarbee et al., 2001). Sedimentary diatom
assemblages *inter alia* can be used to reconstruct past changes in water quality using the
ecological indicator information for each species. Where suitable modern diatom–water
quality calibration data sets exist transfer functions can be generated to reconstruct these
changes. However, in sediment records where diatom diversity is low and affinities of some
species are not firmly established, placing diatom taxa into ecological/environmental
preference groups using literature attributions and field experience can be used to generate
ratio scores relevant to past conditions. The 1$^{st}$ Pond assemblages are suitable for such an
approach, particularly for inferring changes in habitat and water acidity. The acid diatom
index ratio is derived from the sum of acid water indicating taxa comprising *Aulacoseira*,
*Frustulia*, *Pinnularia* and *Eunotia* compared to that of the fragilarioid tychoplanktonic taxa.
Proportions of acidity tolerant to acidity intolerant diatom taxa indicate water pH, total
tychoplankton (temporary phytoplankton) vs. total benthic taxa relate to open water
conditions, subaerial/terrestrial taxa vs. the total assemblage indicate wetland development
and/or in-washed material.
**3.9 Biogenic silica analyses**
The 310 samples were analyzed using a wet-alkaline digestion technique (Conley and
Schelske, 2001). Samples were freeze-dried and gently ground prior to analysis.
Approximately 30 mg of sample was digested in 40 ml of a weak base (0.47M $Na_2CO_3$) at





85°C for a total duration of 3 hours. Subsamples of 1 ml were removed after 3 hours and
neutralized with 9 ml of 0.021 M HCl. Dissolved Si concentrations were measured with a
continuous flow analyzer applying the automated Molybdate Blue Method (Grasshoff et al.,
1983). Biogenic silica content in lake sediments is a proxy for lake productivity.

**3.10 Lipid biomarker and compound specific hydrogen isotopic analyses**

The hydrogen isotopic composition (δ notation) of *n*-alkanes was analyzed by gas
chromatography–isotope ratio monitoring–mass spectrometry (GC-IRMS) using a Thermo
Finnigan Delta V mass spectrometer interfaced with a Thermo Trace GC 2000 using a GC
Isolink II and Conflo IV system. Helium was used as a carrier gas at constant flow mode and
the compounds separated on a Zebron ZB-5HT Inferno GC column (30 m x 0.25 mm x
0.25μm). Lipid extraction was performed on freeze-dried samples by sonication with a
mixture of dichloromethane and methanol (DCM-MeOH 9:1 v/v) for 20 minutes and
subsequent centrifugation. The process was repeated three times and supernatants were
combined. Aliphatic hydrocarbon fractions were isolated from the total lipid extract using
silica gel columns (5% deactivated) that were first eluted with pure hexane (F1) and
subsequently with a mixture of DCM-MeOH (1:1 v/v) to obtain a polar fraction (F2). A
saturated hydrocarbon fraction was obtained by eluting the F1 fraction through 10% $AgNO_3$-
$SiO_2$ silica gel using pure hexane as eluent. The saturated hydrocarbon fractions were
analyzed by gas chromatography – mass spectrometry for identification and quantification,
using a Shimadzu GCMS-QP2010 Ultra. $C_{21}$ to $C_{33}$ *n*-alkanes were identified based on mass
spectra from the literature and retention times. The concentrations of individual compounds
were determined using a calibration curve made using mixtures of C21-C40 alkanes of known
concentration. More details about the GC-IRMS method, including GC oven temperature
program, instrument performance and reference gases used, are given in Yamoah et al. (2016).
The average standard deviation for δD values was 5‰. Due to low sea levels during the time



period of our proxies the δD values of the *n*-alkanes were ice volume corrected (Tierney and
deMenocal, 2013), $\delta D_{corr} = (\delta D_{wax}+1000)/(\delta O^{18}_w*8*0.001+1)-1000$, with interpolated ocean
water $\delta O^{18}_w$ values (Waelbroeck et al., 2002).

Isoprenoid and branched glycerol dialkal glycerol tetraethers (GDGTs) were

measured on the F2 fractions after filtration through 0.45 μm PTFE filters and reconstitution
into a known volume of methanol. Analysis was done using a Thermo-Dionex HPLC
connected to a Thermo Scientific TSQ quantum access triple quadrupole mass spectrometer,
using an APCI interface. Chromatographic separation was achieved using a reverse phase
method similar to the one used by Zhu et al. (2013). Partially co-eluting GDGT isomers were
integrated as one peak in order to obtain data comparable to the normal phase method that has
been in use by the community since Weijers et al. (2007).

One prerequisite for the valid use of brGDGTs is a relatively high branched-

over-isoprenoid tetraether (BIT) index, which was 1.00 throughout the core. Reconstructed
pH values, based on the CBT ratio (Weijers et al., 2007) were stable at 6.6 ±0.1 over the
length of the core, which means that temperature is the dominant environmental factor exerted
on the brGDGT distribution. At the time of measurement, we had not adopted the new method
which separates between 5-methyl and 6-methyl branched GDGTs (De Jonge et al., 2014). As
a consequence, we do not have individual quantifications of 5-methyl and 6-methyl branched
GDGT isomers needed to use the revised $MBT_{5me}$ temperature proxy for mineral soils (De
Jonge et al., 2014) or peat (Naafs et al., 2017), which gives lower RMSE than the original
terrestrial (soil) calibration (Weijers et al., 2007). However, since our data are from lake
sediments, we argue that GDGT-based temperature proxy calibrations based on lake surveys
is a valid approach. Indeed, using the original temperature calibration of Weijers et al. (2007)
resulted in very low temperatures between 0 and 6°C, a cold bias observed in other studies
from lakes. This bias is probably due to the addition of *in situ* produced brGDGTs on top of



any brGDGTs eroded from land (Loomis et al., 2012; Pearson et al., 2011). We therefore used
the global calibration of Pearson et al. (2011), based on a global lacustrine data set and using
mean summer temperatures, including samples from South Georgia Island in the S. Atlantic.
In addition, we also use a calibration based on a large data set of East African lakes from
different altitudes (Loomis et al., 2012), using mean annual temperatures, and which is also
applicable outside of East Africa (Loomis et al., 2012).
**3.11 Calculation of insolation values**
A long term numerical solution for Earth´s insolation quantities (Laskar et al., 2004) was used
for the insolation values, 37-18 ka at 37°S, and calculated with the Analyseries program.
While the austral winter values (W/m2) were based on mean daily June-August insolation
(W/m2), the mean austral summer values were based on the mean daily December-
February insolation.
**3.12 Isotope model simulation**
The isotope model analysis is based on a 1200-year simulation using the isotope enabled
version of the ECHAM5/MPIOM earth system model (Werner et al., 2016) run with natural
and anthropogenic forcings for 800 to 2000 CE (Sjolte et al., 2018). Horizontal resolution of
the atmosphere is 3.75° x 3.75° (T31) with 19 vertical layers, while the ocean has a horizontal
resolution of 3° x 1.8° with 40 vertical layers. Since both the present day situation and our
Nightingale Island record show a continuous impact from the westerlies we deem it valid to
use this late Holocene simulation as an analogue for interpreting the variability of the
westerlies during the time period of study. The outcome of the simulation is presented in the
result section, but further investigation of the model run shows that the multi-decadal
variability of δD at TdC is related to the phase of the Antarctic Annular Mode, indicating that
isotopic variability at TdC is sensitive to large scale SH climate variability (Fig. S4).



### 3.13 Principal component analysis (PCA)

PCA was performed with 14-16 of our variables (proxies) that we expect to respond to

hydroclimate changes, using the C2 program (Juggins, 2007). The aim was to display the

impact of different combination of proxies on samples in a biplot. Our two temperature

proxies MAT and the MST/MAT ratio were both included and excluded in the analyses and

this resulted in almost identical bi-plots in terms of positions of the variables. When the

temperature proxies are excluded PC1 is slightly weaker (38.1 vs 40.9%) while PC2 is

slightly stronger (13.4 vs 11.8%) than when they are included. We therefore include the

temperature proxies in our PCA to illustrate temperature together with the other proxies. All

the variables were centered and standardized before calculation.

## 4. Results

### 4. 1 An island record of glacial climate in central South Atlantic

Thirty-nine 1 m long overlapping cores were taken in February 2010 from three over-grown

crater lakes (Fig. 1C) between lava ridges (Anker Björk et al., 2011). 1P was exceptional in

that it was the only site where sediments older than 18.6 ka were recovered. At 1P the 16.2-

18.6 ka hiatus (Ljung et al., 2015) is marked by a thin silt lamina at 618.8 cm. We retrieved

five overlapping cores below the hiatus with 318.2 cm of sediments before coring was

obstructed at 937 cm by suspected bedrock or boulders. These cores were correlated by

lithology and magnetic susceptibility (MS). The lower 162 cm consist of a dark brown

slightly silty gyttja, overlain by a grey brown silty clay gyttja, all deposited under anaerobic

conditions. Because of the low concentration of plant macro-fossil remains our chronology is

based on 41 [14]C dates of 1 cm thick bulk sediment samples between 620 and 936 cm (Table

S1). Comparisons of [14]C dates of bulk sediment and plant remains (wood and peat) have

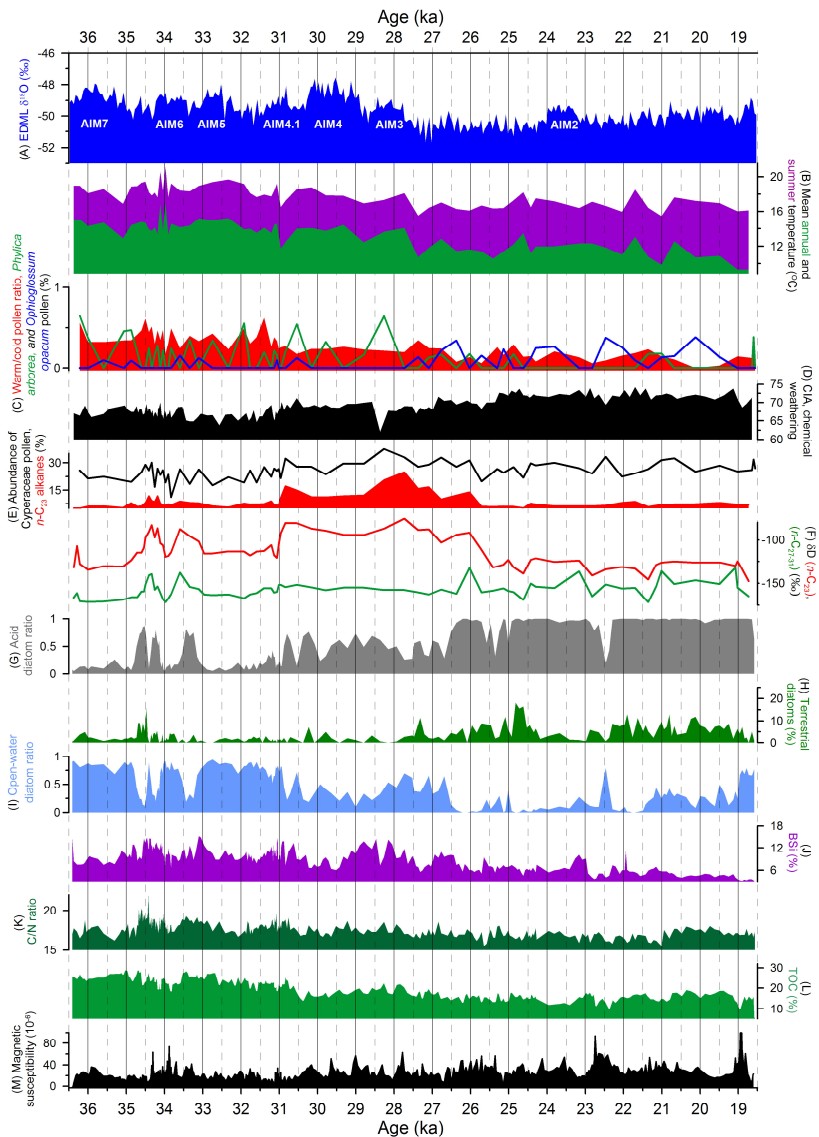

**Figure 4.** Antarctic ice core data and some of the proxy data from the sediments in 1st Pond, between 36.4 and 18.6 ka. (A) The EDML $\delta^{18}$O record (EPICA Community Members, 2006) showing AIM 7-2. (B) Mean annual and summer temperature from the GDGT analyses and calibrated with Pearson et al. (2011) and Loomis et al. (2012), respectively. (C) Warm pollen ratios, % *P. arborea* pollen and % *Ophioglossum* spores. (D) Modified chemical index of alteration (CIA). (E) % Cyperaceae pollen and *n*-$C_{23}$ alkanes. (F) $\delta$D values (‰) of *n*-$C_{23}$ and *n*-$C_{27-31}$ alkanes. (G) Acid diatom ratios. (H) % terrestrial diatoms. (I) Open water diatom ratios. (J) % biogenic silica (BSi). (K). C/N ratios. (L) % total organic carbon (TOC). (M) Magnetic susceptibility (MS) expressed as $10^{-6}$ SI units. All proxies are related to the age scale on the x-axes.



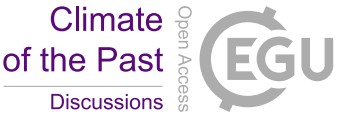

shown good concordance (Ljung et al., 2015; Ljung and Björck, 2007). Our age model (Fig.
3) displays a mean sedimentation rate of 0.18 mm yr$^{-1}$, but with considerable variation.

A large set of proxy data from 1P (Figs. 4-5, Figs. S1-S3, Table S2) was

analyzed and provides information about local changes such as soil conditions/erosion,
weathering, vegetation composition, organic productivity, lake conditions and P/E ratios.
Other proxies (GDGTs) display regional changes in temperature such as mean annual (MAT)
and mean summer temperatures (MST) and hydroclimate conditions (deuterium isotopes),
such as the source water of terrestrial and aquatic plants including evaporative conditions.
Principal component analysis (PCA) was performed to investigate co-variability between
proxies, showing the interplay of changes in hydroclimate driven by oceanic and atmospheric
circulation changes (Fig. 6).

In agreement with the supposed minimum age of pond formation through

volcanic activity (Anker Björk et al., 2011), the bottom of 1P has an age of 36.4±0.3 ka. Our
temperature records (Fig. 4B) show an oscillating pattern, with the largest change at 27.5 ka,
and share many similarities with the EDML curve (Fig. 4A). Before 27.5 ka MAT and MST
vary between 17-12°C and 21-17°C, respectively, while the variation is between 13-9°C and
18.5-15.5°C, respectively, after 27.5 ka. In terms of pollen as a local temperature indicator it
is known that *Phylica arborea*, *Acaena sarmentosa* and two Asteracea plant types are
sensitive to cold conditions (Ryan, 2007). They make up warm pollen types at NI and the
warm/cold pollen-types ratio (Fig. 4C) shows large variations until 31.4 ka, followed by a
two-step decline largely in contrast to the spore abundance of the cold tolerant *Ophioglossum*
*opacum* fern, and with a trend similar to the temperature curves. In comparison to Holocene
sediments from NI (Ljung and Björck, 2007), the glacial pollen record from 1P (Fig. 5) shows
less variability, and the most distinct difference is the very low abundance of the only tree
species pollen on the island, the frost limited *P. arborea*. Based on lapse rates, with 65-130 m



lower sea levels 35-18 ka (Lambeck et al., 2014), and today´s distribution of *P. arborea* on
TdC and Gough Island (Ryan, 2007) we can estimate that its absence after 28 ka implies
minimum winter temperatures at least 3°C lower than today, which agrees well with our MAT
curve (Fig. 3B).

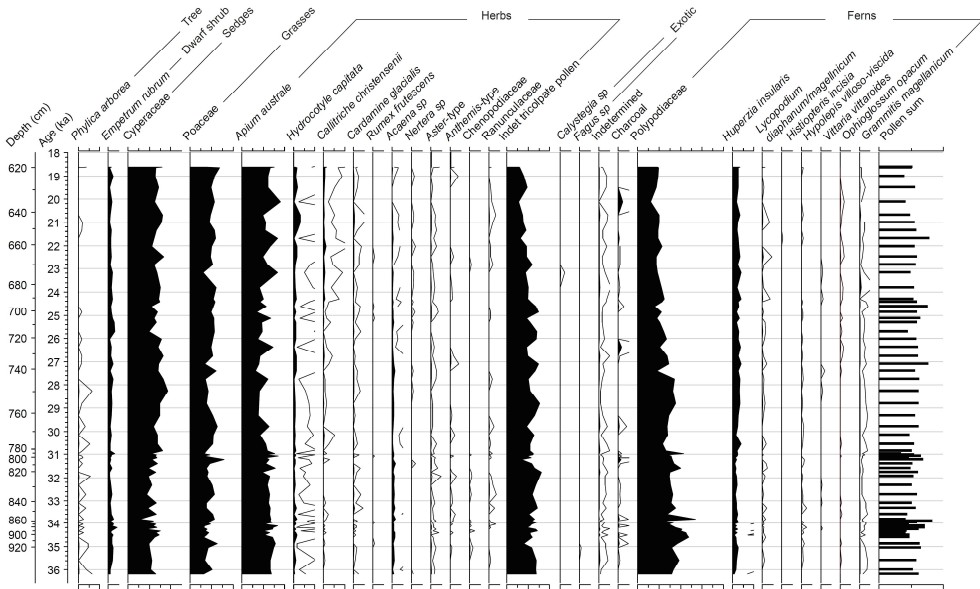

**Figure 5**. Pollen diagram from 1ˢᵗ Pond, Nightingale Island. The diagram shows relative abundance (%)
of the pollen taxa. Note that it is both related to depth (cm) and age (ka) on the y-axis, the latter according
to the age-depth model in Fig. 3.

To evaluate changes in the degree of weathered material we used a modified

Chemical Index of Alteration (CIA) (Fig. 4D). The long-term development can be divided
into three phases with initially low but very variable values until 31 ka, a second phase with
stable intermediate CIA values until 27 ka, followed by higher and varying values (Fig. 4D).
Magnetic susceptibility (MS) shows centennial-millennial oscillations superimposed on an
increasing trend from the bottom to the top of the core (Fig. 4M), and is an indicator of in-
washed mineral matter from magnetite-rich basaltic rocks of the catchment. The values of



total organic carbon (TOC) and biogenic silica (BSi) (Figs. 4L and J) reflect organic and
aquatic productivity in and around the lake with highest values in the oldest section. TOC
shows a general decline and BSi oscillates with higher values until 28 ka, after which it
gradually drops. The fairly high C/N ratios (Fig. 4K), with a mean value of 17.6, show that
organic matter is a mix of terrestrial and aquatic sources. The high and oscillating ratios in the
older section followed by a gradual decline implies terrestrial sources dominating until 28 ka,
after which time aquatic sources become more important. With respect to stable isotopes (Fig.
S1), the high $\delta^{15}N$ values imply a marine origin possibly related to presence of marine birds,
such as Great Shearwater and Albatrosses which have a great impact on the Ponds today,
suggesting a more or less continuous impact of SHW.  Rising $\delta^{13}C$ values at 25.7 ka are
consistent with the declining C/N ratios after 28 ka, i.e. more aquatic material with enriched
$^{13}C$, and possibly higher influence from $C_4$ grasses.

Unlike the pollen record (Fig. 5), the diatom record shows large shifts and the 33

diatom taxa (Fig. S2) have been classified into three environmental forms. Changes in these
groups imply shifts in aquatic and environmental conditions in and around the lake. They
show a lake with open water early in the record, followed by shifting lake levels between 35-
33 ka (Fig. 4I), supported by $\delta D$ values of long- and mid-chain *n*-alkanes (Fig. 4F). At 31 ka
the open water ratios drop and reach a minimum at 29 ka, in anti-phase with the acid water
diatom ratios (Fig. 4G), followed by a rise until 26.6 ka. Thereafter acid species dominate, as
oligotrophic wetland encroached around the lake, while periods of more terrestrial diatoms
imply episodes of in-washed diatoms from the surroundings. Around 21.2 ka more open water
conditions prevail again with high ratios 19-18.6 ka, before the lake dried out (Ljung et al.,
2015). The shifts in diatom communities shows that 1P went through substantial hydrologic
changes, some of which were rapid, induced by changing P/E ratios, in contrast to the fairly
stable vegetation around the lake as seen in the pollen record.



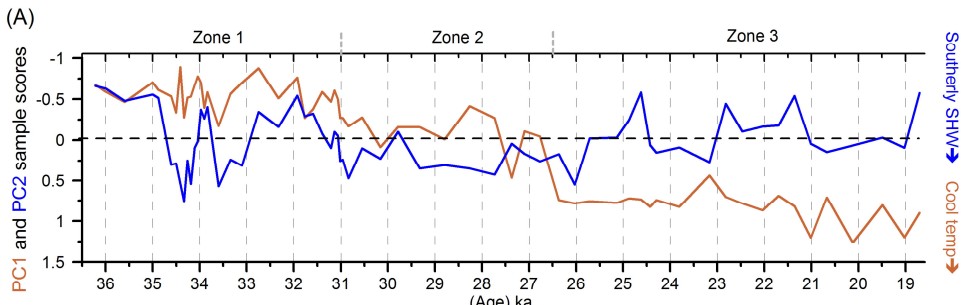

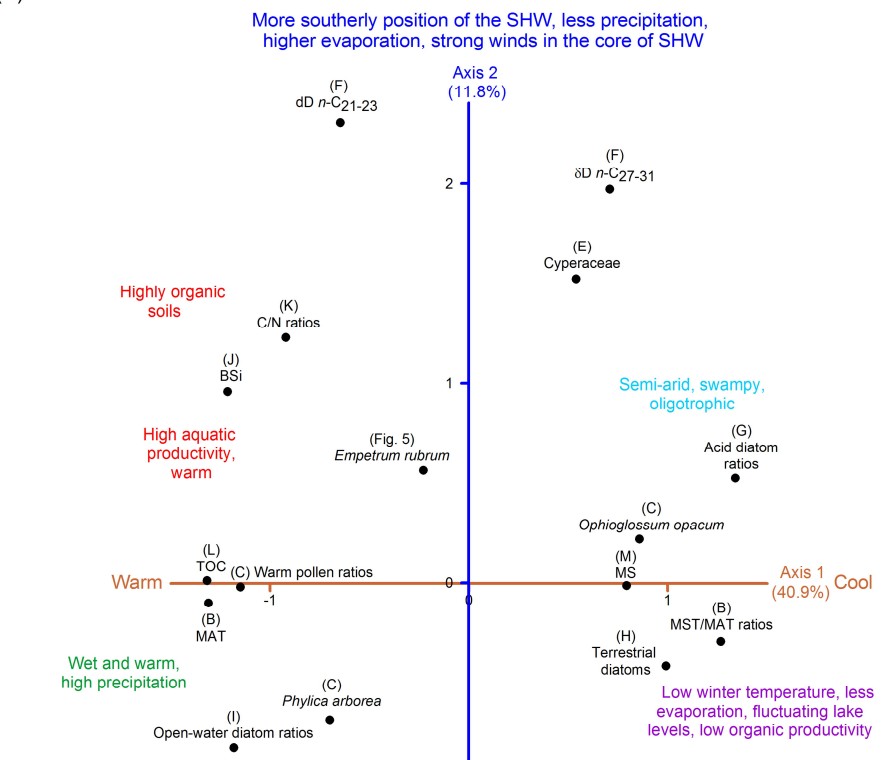

**Figure 6.** Principal component analysis (PCA) of 16 proxies from $1^{st}$ Pond. (A) Scores of the first two principal components related to age and the three PCA zones. Note that negative values point upwards and how PC1 and PC2 values relate to temperatures and SHW to the right y-axis. (B) PCA plot shows the loadings of the 16 proxies (shown as black dots and black text with reference to proxies in Fig. 4, except for *Empetrum rubrum*). PC1 (red brown) and PC2 (blue) accounts for 40.9 and 11.8% of the variance, respectively. The interpretations of the two axes are shown by red brown and blue texts, and the interpretations of the four segments are based on the combined positions of the proxies in the plot, and are shown in four different colors. MST/MAT=mean summer temperature/mean annual temperature, i.e. a proxy for low winter temperatures.



### 4.2 Linking **the Nightingale Island record to South Atlantic hydroclimate**

The hydrological sensitivity of a basin like 1P makes it ideal to place local changes into the

context of regional hydroclimate shifts. To analyze the variability through time Principal

Component Analysis (PCA) was carried out on a data set with 16 hydroclimate-sensitive

proxies resulting in 3 PCA zones (Fig. 6A). Note that resolution of the PCA record depends

on the proxy with least common sample levels (Table S2), in this case biomarker analyzes.

Therefore the temporal resolution of the PCA is not as high as some ice core and marine

records. Based on the proxy loadings in the PCA plot (Fig. 6B), it can be divided into four

different segments with variable hydroclimate and environmental conditions. The importance

of temperature proxies on Axis 1 (40.9% of the variance) is obvious where reconstructed

MAT, warm pollen ratios, *Phylica arborea* pollen, BSi, TOC and open-water diatoms show

warm humid conditions to the left (negative) in the biplot (Fig. 6B), vs cooler and drier to the

right. The latter is accentuated by *Ophioglossum* spores, a fern growing at high and cold

altitudes on TdC, and colder winter temperatures implied by the MST/MAT ratio. Axis 2

(11.8% of the variance) is linked to hydrologic indicators being dominated by the δD values

of the aquatic $n$-$C_{21\text{-}23}$ and terrestrial $n$-$C_{27\text{-}31}$ alkanes (Fig. 6B). We interpret higher δD values

(positive axis 2 values) to show stronger influence of more local air masses, with more

evaporation and semi-arid conditions, also shown by *Empetrum rubrum* pollen in the upper

left quadrant, while the upper right quadrant of the plot shows an acid oligotrophic swampy

setting. The segment to the lower right in Figure 6B displays cold conditions and in-wash of

terrestrial diatoms as an effect of higher lake level during episodes of more precipitation. The

lower left represents warm and wet conditions, implied by *P. arborea* pollen and open water

diatoms, and in general, negative axis 2 values relate to more negative δD values.

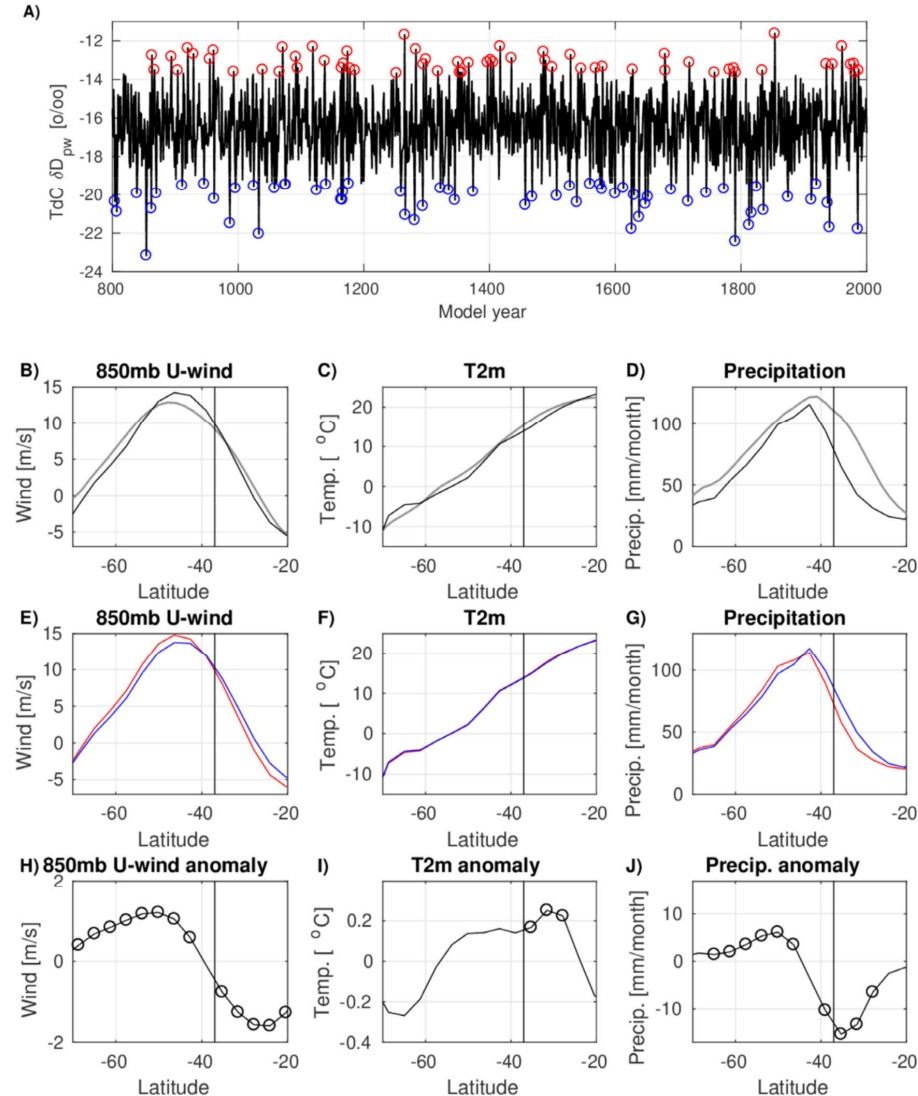

**Figure 7.** Zonal mean changes in wind, temperature and precipitation related to δD variability at TdC.
A) Time series of simulated precipitation weighted annual mean δD at TdC, with values above and
below the 95[th] percentile indicated with red and blue circles, respectively. This selection of high and low
δD is used to define the data in figures E-G. (B-D) Annual modeled (black) South Atlantic zonal mean
(30°W to 0°W) westerly wind speed (850mb U-wind, positive towards east), 2m temperature (t2m) and
precipitation compared to the 20[th] Century Reanalysis climatology 1981-2010 (Compo et al., 2011). (E-
G) Composites of annual modeled zonal mean (30°W to 0°W) westerly wind speed (850mb U-wind),
2m temperature (t2m) and precipitation for high (red) and low (blue) δD at TdC defined in (A). (H-J)
High-minus-low anomalies of model output are shown in (E-F). Circles indicate significant anomalies
(p < 0.01) calculated using two-tailed Student's t-test. The vertical bars in (B-J) show the latitude of NI
at 37°S.



Observations of the isotopic content of precipitation are very sparse around TdC.
Therefore we have investigated the hydroclimate variability with an isotope enabled climate
model. To illustrate the relation between the position of the westerlies and the isotopic
composition of precipitation at TdC in the simulation, we selected extreme values of high and
low δD at TdC (Fig. 7A), and made composite anomalies of the annual mean westerly wind
strength at 850mb (u850mb, Fig. 7A), precipitation (Fig. 7B), 2m temperature (t2m, Fig. 7C)
and precipitation weighted δD (Fig. 7D) for high-minus-low δD at TdC. This shows that the
variability of δD in precipitation at TdC is only weakly dependent on local temperature.
Instead, shifts in δD at TdC are related to large scale changes in precipitation and the position
of the westerlies. Positive δD anomalies at TdC imply a more southern position of the core of
the westerlies with drier conditions at TdC, and negative δD anomalies at TdC denote a more
northern position of the core of the westerlies bringing more polar air masses with wetter
conditions at TdC. From Figs. 7 and 8, we note that the shifts in TdC precipitation are
governed by the precipitation zone on the northern flank of the westerlies shifting with the
position of the westerlies themselves. We therefore conclude that our model analysis shows
that isotope variability in precipitation at TdC is mainly related to shifts in large scale
circulation. High δD values at TdC imply a more southerly SHW position with stronger winds
in its core, while low δD values show a more northerly SHW position with weaker winds
(Figs. 7E and H). Our analysis also shows that high (low) δD values are related to less (more)
precipitation at TdC, but shows little dependency on temperature (Figs. 7F and I, and 8C).
The modelled relationship between δD and precipitation corresponds well to the PC2
variability of the proxies (Fig. 6B); for example high PC2 and δD values relate to more
Cyperaceae (lake overgrowth) and *Empetrum* pollen values (arid soils) and more acid diatoms
(swampy), while low PC2 values relate to open-water (lake) and terrestrial (flushed-in)
diatoms.

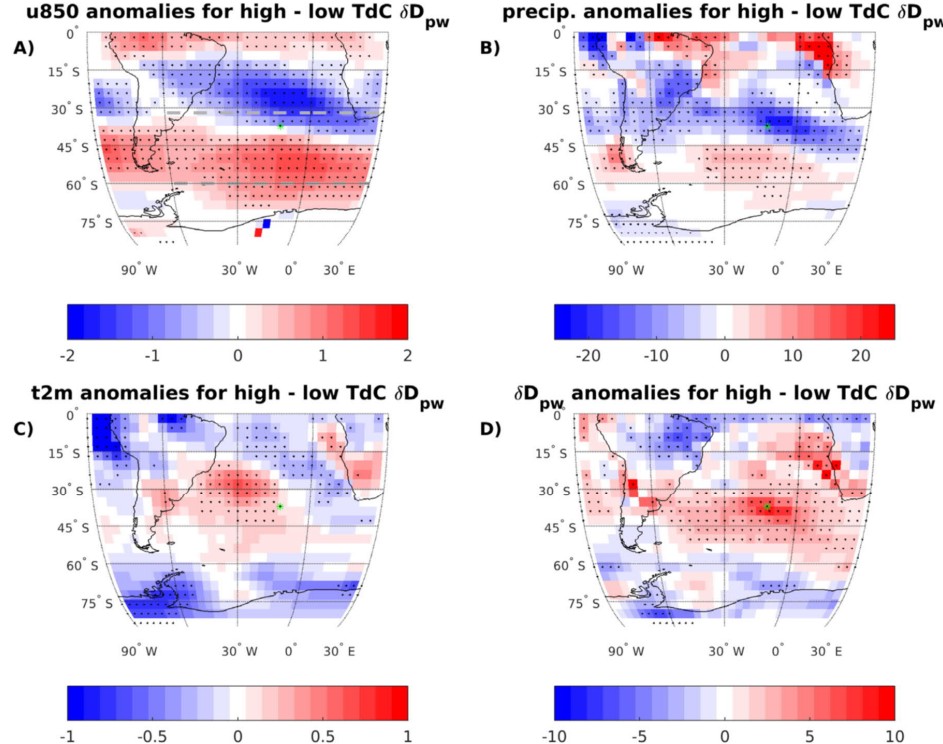

**Figure 8.** Composite maps of changes in wind, precipitation, temperature and δD related to δD variability at TdC, showing annual anomalies based on composites for high and low δD at TdC (see Figure 7A). A) Westerly wind speed (850mb U-wind, positive towards east, [m/s]). The dashed gray lines show the approximate northern and southern boundaries of the westerlies (850mb U-wind > 5 m/s) to clarify that high TdC δD is related to a southward shift in the westerlies. B) Precipitation [mm/month]. C) 2m temperature (t2m, [ºC]). D) precipitation weighted δD [‰]. Stippling indicates significant anomalies (p < 0.01) calculated using a two-tailed Student's t-test. The green spot shows the position of TdC.

## 5 Hydroclimate correlations and interpretations
### 5.1 The large-scale hydroclimate pattern

The three PCA zones displayed in Figures 6A and 9, dated to 36.2-31.0, 31.0-26.5 and 26.5-

18.6 ka, show a trend and pattern which is recognizable in much of our data set as well as in

the EDML (Fig. 4A) and South Atlantic marine record (Fig. 10B). Zone 1 is fairly warm but

oscillates between low and high PC2 values, related to more northerly and weaker SHW, and

more local air masses with stronger westerlies in a more southern position, respectively. Zone





2 is generally more stable with some minor oscillations with more southerly SHW and
corresponds largely to the fairly warm period in Antarctica with the three isotope maxima
AIM4.1, AIM4 and AIM3 (Fig. 4A), and a stable and mild period in the South Atlantic marine
realm (Fig. 10B). Zone 3 shows a cooling trend, also visible in the EDML and marine record,
with variable SHWs. It appears that TdC was continuously influenced by the SHW, as shown
by the absence of arid conditions and generally low $\delta$D values, verified by humid conditions
in southwestern-most Africa throughout most of MIS3 and MIS2 (Chase and Meadows,
2007). Apart from the resemblance between the long-term trends in Antarctic ice core data
and marine data at 41°S in the South Atlantic (Barker and Diz, 2014) with our data it is, in
spite of our lower resolution, interesting to compare our PC2 and $\delta$Dn-C$_{C27-C31}$ records (Figs.
4A and 10G) with other regional records related to SHWs. Taking age uncertainties of a few
hundred years into account we note a resemblance with marine Fe fluxes at 42°S (Martínez-
García et al., 2014) where low $\delta$D values (Fig. 10G) co-vary with high Fe fluxes (Fig. 10F)
due to northerly SHW in a cooler Southern Hemisphere, thus expanding the Patagonian dust
source. Similar co-variability can be seen in the $\delta^{18}$O record on fluid inclusions of SE
Brazilian speleothems (Millo et al., 2017) where low values (Fig. 10E) imply strengthening of
the monsoon shifting the South Atlantic atmospheric system southwards, including SHW. We
also note that the Antarctic CO$_2$ record (Fig. 10C) and the [CO$_3^{2-}$] record (Gottschalk et al.,
2015) from the South Atlantic (Fig. 10D), inferring AMOC strength and Southern Ocean
ventilation, share similarities with our SHW records, as described in the section below.



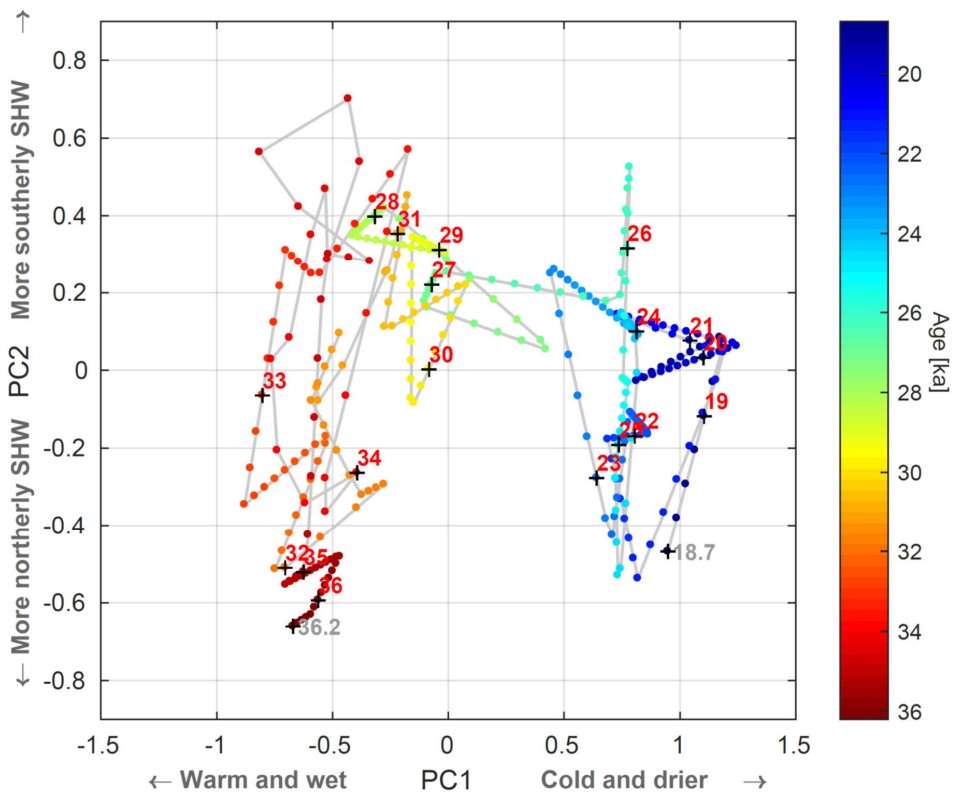

**Figure 9.** Parametric plot of the PC1 and PC2 sample values as a function of time shown by the color bar to the right. Red numbers denote each ka with grey numbers at the start and end of the plot. Data was interpolated to 50-year time steps to illustrate rate of change; the larger distance between dots the more rapid change. Note that the hydroclimate interpretations from Figure 6B are shown on the two PC axes.

## 5.2 A detailed hydroclimate scenario for the central South Atlantic

Due to chronological uncertainties in all records, lower resolution in some records and the

complex phase-relationships during abrupt interhemispheric climate shifts (Markle et al.,

2016), detailed comparison of short-term variations across sites has to be treated with caution.

In spite of these short-comings we will present a scenario based on our record and likely

correlations.

The start of our record shows warm and wet conditions with northerly SHW,

coinciding with the long and warm AIM7 followed by a cooling (Fig. 10J) at the onset of





DO7. This is followed by the very dynamic period 35-33 ka, shown by high sedimentation
rates (Fig. 3) and peak variability in terms of both rapidity and amplitude (Fig. 9).  Such
variability is also seen in marine and ice core records, and in spite of the age uncertainties at
34-35 ka (Fig. 3) we tentatively correlate this period in our record to the end of DO7 and the
minimum between AIM6 and AIM7. This corroborates the overlaps and time lags that have
between postulated for DO and AIM events (Markle et al., 2016; Pedro et al., 2018; WAIS
Divide Project Members, 2015). At 34 ka we note a temperature peak at the onset of AIM6
(Figs. 4 and 11) followed by falling temperatures, $\delta D$, Fe flux and $CO_2$ values and high
humidity (Figs. 10G, F, C and H). This change reflects northerly and weaker westerlies, with
rising speleothem $\delta^{18}O$ and WAIS $d_{ln}$ values (Figs. 10E and 11E), denoting the start of DO6
with a warming of the NH (Fig. 10A). This caused northwards shifting ITCZ and SHW in line
with the theory that the atmospheric circulation system moves towards the warmer
hemisphere, responding to the change in the cross-equatorial temperature gradient (McGee et
al., 2014). At 33.5 ka we see a southward SHW shift with rising temperatures and higher $CO_2$
and lower WAIS $d_{ln}$ values with dry conditions. We relate this to the  onset of AIM5; a
warming which is interrupted at 32.8 ka by a northerly SHW shift and wetter conditions (Figs.
11F and 10H) possibly triggered by DO5. This partly continues until 31.7 ka when SHW
moves south with a minor temperature rise (Fig. 11D) and decreasing humidity, possibly as a
response to the post-DO5 cooling (Fig. 10A).  The high variability and large amplitude of the
changes of Zone 1 (Fig. 9) have facilitated conceivable correlations to other records. Based on
these we can conclude that at large, PC2 implies northerly shift of the SHW during warm
North Atlantic periods, and a more southerly position during warm periods in Antarctica, also
in line with interpretation of Antarctic deuterium excess data (Markle et al., 2016).

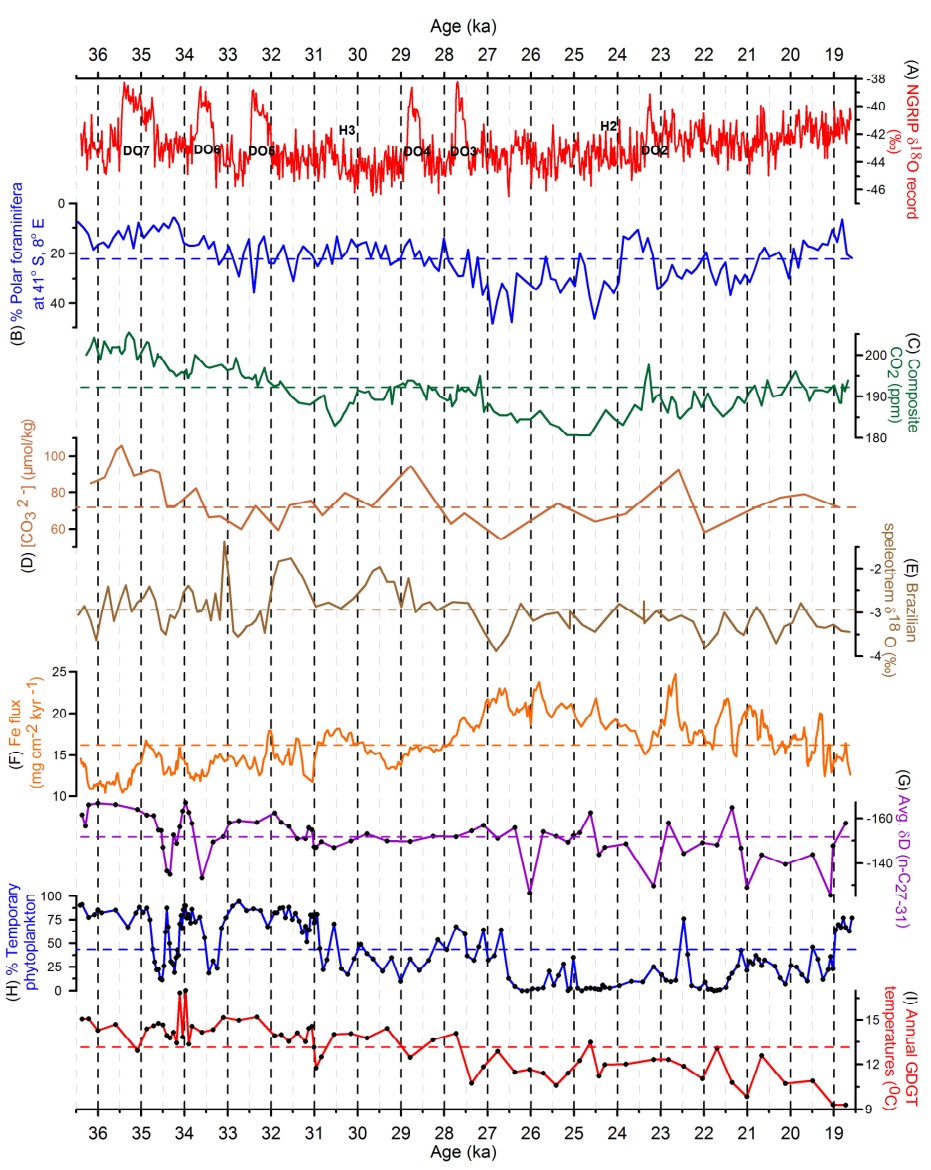

**Figure 10.** Comparisons between other proxy records (A-F) and Nightingale Island proxies for SHW (G), wetness (H) and temperature (I), with mean values as broken lines. (A) $\delta^{18}O$ values from the NGRIP ice core (Andersen et al., 2006) showing DO and H events. Ice core records are on a common time scale (Veres et al., 2013). (B) Abundance (%) of polar foraminifera at 41°S in the S Atlantic (Barker and Diz, 2014). (C) Composite Antarctic $CO_2$ record from Siple Dome (Ahn and Brook, 2014) and WAIS (Stenni et al., 2010). (D) $[CO_3^{2-}]$ data at 44°S in the South Atlantic (Gottschalk et al., 2015). (E) Speleothem $^{18}O$ record on fluid inclusions from SE Brazil (Millo et al., 2017). (F) Fe flux data in the South Atlantic at 42°S (Martínez-García et al., 2014). Then follow NI data, (G) Average $\delta D$ values for the terrestrial $n$-$C_{27-31}$ alkanes. (H) Abundance (%) of temporary phytoplanktonic diatoms implying relative water depth. (I) Annual NI temperatures from the GDGT analyses. Note that that sample levels are shown by a dot in (G)-(I) and that y-axes of (B) and (G) show higher values downwards to facilitate comparisons to other proxies.





The Zone 1/Zone 2 boundary at 31 ka (Fig. 6A) is a dynamic transition, shown

by many proxies and peak sedimentation rates (Figs. 9 and 3). The 4.5 ka long and stable
Zone 2 (Fig. 6A) is characterized by fairly high but slightly decreasing temperatures and as in
Zone 1 a dominating southerly SHW position.  It is possible, taking age uncertainties into
account, that H3 at 30.5 ka (Fig. 10A) triggered the southbound SHW and the rising $CO_2$ and
MAT values, and the reduced humidity between 31-30 ka (Figs. 11F, 10C, 10I and 10H). The
following long and warm AIM4 may have stabilized conditions in the South Atlantic in spite
of the DO4 event at 28.8 ka. This stability is also seen in marine records (Fig. 10B), and the
rather stable southern position of the SHW agrees with the fairly high $CO_2$ values between 30-
27.2 ka and with falling and rather low Fe fluxes (Fig. 10F).  We also note higher lake
evaporation from δD values of the aquatic $n$-$C_{23}$ (Fig. 4F), in concert with rising summer
insolation (Fig. 11A). Around 27.5 ka we see a brief response in some of the proxies to the
short DO3 event (Fig. 10A), such as the MAT and PC2 records (Figs. 11D and F) and is also
noticeable in e.g. the marine and Brazilian monsoon records (Figs. 10B and E).

The start of Zone 3 constitutes the most drastic change in our record (Figs. 6A

and 9) but timing varies between proxies (Fig. 4). MAT, TOC and C/N ratios start to decrease
already at 28 -27.5 ka, coinciding with DO3, the biologic proxies (Figs. 4C and G-J, Fig. S2)
respond slightly later possibly because they do not react until certain hydroclimate thresholds
for the vegetation and algae flora are reached. The Zone 2-3 transition is roughly
simultaneous with the onset of LGM in Antarctica (Fig. 10C), when 1P switched from a lake
to a wetland, coinciding with increased abundance of polar foraminifera at 41°S (Fig. 10B).
This may be an effect of the STF moving north of TdC, a meridional shift comparable to what
has been shown from the eastern Pacific (Kaiser et al., 2005). The fairly stable PC1 values
show cool and less humid LGM conditions, while the variable PC2 values imply shifts in the
position of SHW (Fig. 11F). There is also a good correspondence between our δD ($n$-$C_{27-31}$)



maxima after 27 ka and Fe flux minima from the South Atlantic (Figs. 10G-F), both indicating
southerly shifts of SHW. During this period our data also show generally higher mean $\delta D$ (n-
$C_{27-31}$) values than in Zone 1, implying a more southern position of SHW during the Antarctic
LGM, as seen in some modeling results (e.g. Sime et al., 2016). This is also compatible with
the fact that the LGM temperature lowering in the Northern Hemisphere (Johnsen et al., 1995)
was much larger than in the south (Stenni et al., 2010), shifting the atmospheric system to the
south due to changes in the cross-equatorial gradient (McGee et al., 2014), as implied by the
speleothem $\delta^{18}O$ data (Fig. 10E) showing increased precipitation (Millo et al., 2017).

After 26.5 ka we note phases of less humid swampy oligotrophic conditions on

NI at 26, 24.5-23, 22 and 20.5-19 ka (Fig. 10H) interrupted by periods of more or less open
water, possibly driven by shifts of SHW. The former often show enriched $\delta D$ values (Fig.
10G), while the latter were characterized by higher precipitation and more depleted $\delta D$
values. Regarding the response of $CO_2$ to these SHW shifts we note a fairly good agreement
between low/falling $CO_2$ values and a northerly SHW position, and vice versa. For example,
the $CO_2$ minimum at 24.5-25 ka (Fig. 10C) matches with an extreme northern SHW position
(Figs. 10G and 11F), and the $CO_2$ peak at 23.3 ka agrees with the end of a long phase of
southwards moving SHW. The latter might have been triggered by the onset of H2 at 24.1 ka
(Fig. 10A) followed by the inception of AIM2 (Fig. 11C).

The absence of *P. arborea* (Figs. 4C and 5) and our temperature proxies (Fig.

4B) imply that minimum winter temperatures at our site were occasionally below zero,
especially after 26 ka; periods of frost also explain increased weathering (Fig. 4D). Between
23 and 19 ka the Antarctic winter sea ice reached 47°S in the South Atlantic (Gersonde et al.,
2005), only some 1000 km south of TdC. Our 1P record shows a declining temperature trend
during the end of this period (Fig. 10I), in contrast to rising temperatures in Antarctica and
South Atlantic (Figs. 11C and 10B). This regional temperature anomaly may be explained by



the declining summer insolation at the latitude of Tristan da Cunha (Fig. 11A), and may also,
at the end of LGM, be related to break-up of Antarctic ice shelves as sea levels rose, causing
cooler conditions further north. In fact, temperature minima after 19 ka are seen in both our
record and in marine data (Figs. 10I and B), as well as a δD minimum (Fig. 10G).

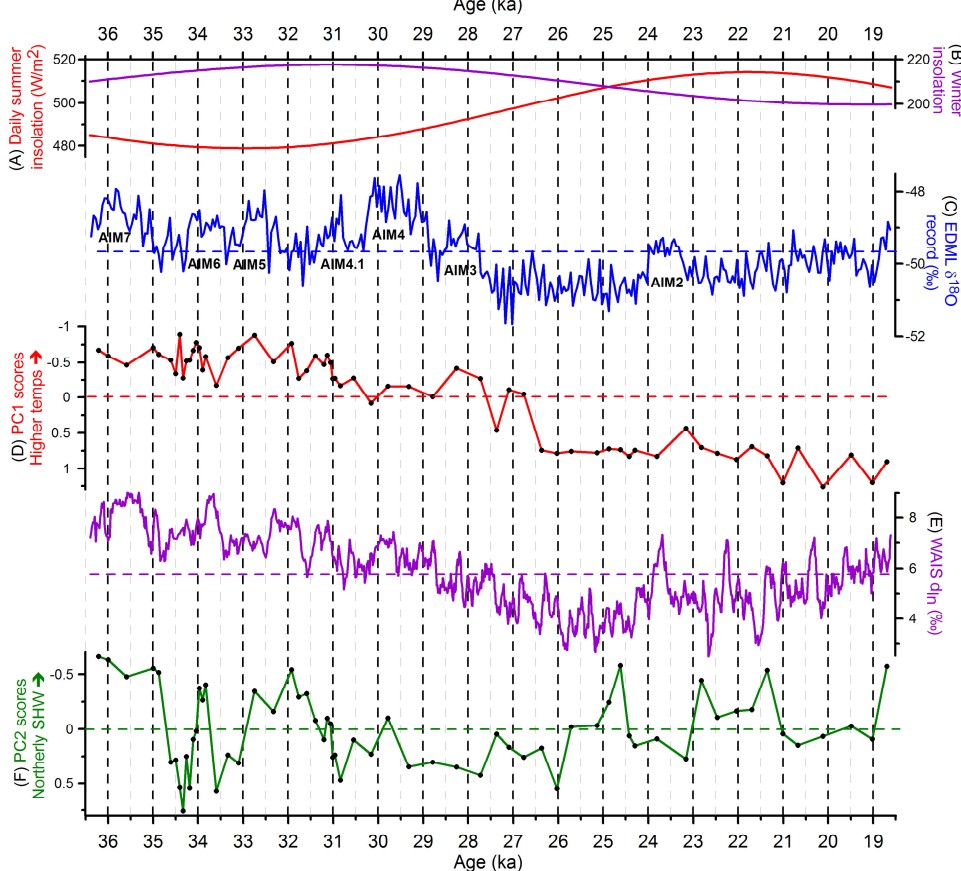

**Figure 11.** Comparison between our PC1 and PC2 records and other relevant data. (A and B) Mean daily
summer and winter insolation at 37°S (Laskar et al., 2004). (C) EDML δ18O record (EPICA Community
Members et al., 2006) with Antarctic Isotope Maxima (AIM). (D) PC1 scores implying temperature
shifts at NI. (E) WAIS $d_{ln}$ values from west Antarctica (Markle et al., 2016). (F) PC2 scores indicate
impact of SHW at NI. Note that sample levels, i.e. time resolution, for the PC records are shown as dots.







### 5.3 A climate synthesis


In general, our data implies two main climate modes for the study period, separated by a
transition period 31-26.5 ka, Zone 2. This is displayed in Figure 9, with pre-LGM (Zone 1)
clearly separated from the LGM period (Zone 3) on Axis 1, but also with higher variability of
the pre-LGM period. This variability is possibly related to an active bipolar seesaw
mechanism during Zone1/MIS3 even at the fairly low latitudes of TdC, triggering N-S shifts
of SHW and related hydroclimate conditions. Any $CO_2$ effects from the rapid SHW shifts in
Zone 1 are not discernible, but the dominating more northern SHW position may have
resulted in the general $CO_2$ decline (Fig. 10C). With the onset of Zone 2 there may be a
stronger link between $CO_2$ and SHWs. In view of  carbon-cycle time lags,  the mainly
southerly positioned and more intense SHW at 31-27.5 ka (Fig. 11F) may have resulted in the
rising and higher $CO_2$ concentrations at 30.5-27.2 ka (Fig. 10C), with more upwelling, $CO_2$
outgassing and less sea ice. The LGM mode is characterized by falling and low temperatures,
lack of clear effects of the bipolar seesaw mechanism, possibly due to the much stronger
cooling in the north as the cross-equatorial gradient changed. The variability is mainly related
to proxies associated with SHW changes, as summarized by PC2, with a similarly high
frequency variability of WAIS $d_{ln}$ and Fe fluxes (Figs. 11E, 10F), with resulting $CO_2$
variability. However, a key difference between our SHW proxies (PC2) and the WAIS $d_{ln}$
record is that the latter represents SHW variability superimposed on large scale temperature
trends while our PC2 record reflects the SHW signal without temperature impact.

Thus, the largest change in our record occurs after 27.5 ka when the effects of

the strong post-DO3 cooling of the Northern Hemisphere start dominating the hydroclimate of
the South Atlantic with highly variable SHW after 25 ka ; possibly a prerequisite for the
oscillating $CO_2$ levels after the $CO_2$ minimum at 25 ka (Fig. 10C).





**6.** Conclusions
Our 1P data, reflecting terrestrial and aquatic responses to shifting atmospheric conditions,
show that the glacial hydroclimate of South Atlantic mid-latitudes experienced varying
degrees of humidity, but with more or less continuous impact of SHW. Temperature
conditions were in general warm but oscillating during MIS3, with shifting strength and
positions of the westerlies. Weaker and northwards moving SHW at the onset of NH
interstadials with stronger and southerly westerlies during NH stadials partly reflect the
complex processes behind phase relationships between Greenland and Antarctic ice core
climate records (Pedro et al., 2018). These shifts, possibly triggered by changes in the cross-
equatorial gradient, are to some extent manifested by rising (falling) $CO_2$ levels when SHW
was stronger (weaker) and located more towards the south (north), in line with Holocene
records (Saunders et al., 2018). The largest variability in our record is seen during the fairly
warm and humid period 36.5-31 ka with frequent and abrupt shifts, followed by a fairly stable
period 31-27 ka with slowly declining temperatures and dominating southerly SHWs. The
largest over-all change occurs after 27 ka , exhibited by a distinct cooling trend. This early
mid-latitude cooling is in phase with LGM in Antarctica, consistent with some modeling
results (Fogwill et al., 2015). We think this represents a mode shift in hydroclimate; from the
highly variable MIS3 conditions through the more steady conditions 31-27 ka (Figs. 11D and
F) into LGM with its cool and less humid climate, perhaps as a result of the SF moving north
of TdC. The variable position of SHW (Fig. 11F), with particularly high δD values at 26, 23.1,
21 and 19.1 ka (Fig. 10G), is noteworthy, inferring fairly sudden and distinct southerly shifts
of the westerlies. The end of our record shows that cool conditions persisted in these SH mid-
latitudes until at least 18.6 ka. This might have been a combined effect of declining summer
insolation and northward shifting westerlies (Figs. 11A and F), conveying cold air masses, sea
ice and ice bergs far north from collapsing Antarctic ice shelves (Weber et al., 2014).



**Author contributions.** S.B. was the initiator of the study, received funds, drilled and described cores, carried out sampling and XRF analyses and contributed with most writing, J.S. contributed with interpreting data, much writing, ran the isotope model experiment (ECHAM5-wiso/MPI-OM) and analyzed all modeling results, K.L. drilled and described cores, carried out sampling, analyzed C, N, $^{13}$C, $^{15}$N, pollen and contributed with writing, F.A contributed with the age model and some writing, R.F. contributed with interpreting and analyzing diatom results and some writing, R.H.S. helped interpret biomarkers and hydrogen isotopes and contributed with some writing, M.E.K. analyzed XRF results and contributed with some writing, T.F.S. contributed with creative inputs and some writing, S.H. sampled and carried out diatom analyzes, H.J. carried out multivariate statistics, Y.K.K.A. analyzed biomarkers and hydrogen isotopes, R.M. calculated insolation values and contributed with little writing, J.E.R. carried out biomarker analyses and calibrated the GDGTs and N.V.d.P, carried out biogenic silica analysis. All commented on the manuscript.

**Acknowledgements.** The co- members of the 2010 Tristan expedition (M. Björck, A. Björk, A. Cronholm, J. Haile, M. Grignon) and Tristan islanders are gratefully acknowledged for hard work at sea and on Nightingale I. The isotope enabled climate model, ECHAM5-wiso/MPI-OM, was run at the AWI Computer and Data Center. We thank M. Werner for helping to set up and run the model simulations, S. Barker, F. Cruz and C. Millo for providing us with their data, G. Ahlberg for pollen sample preparations and Å. Wallin for magnetic susceptibility measurements. We are grateful for financial support, incl. expedition costs, from the Swedish Research Council (VR), the Crafoord Foundation, the Royal Fysiographic Society, the LUCCI Centre in Lund and the Lund and Stockholm universities. We dedicate this paper to Charles T. Porter, our skipper on his ketch Ocean Tramp, who challenged all kind of weather in the South Atlantic to retrieve our unique sediment cores. However, he sadly died suddenly in March 2014 while preparing for our next expedition: a great loss in many respects but mostly as an invaluable, memorable friend and colleague.





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
