# Peer review of "A SOUTH ATLANTIC ISLAND RECORD UNCOVERS SHIFTS IN 1"

_Climate of the Past, 2019_

## Referee Comment (RC1) · Anonymous Referee #1 · 4 Jul 2019

General comments: BjoÌĹrck et al. generated a multi-proxy record from a sediment core in a mid-latitude island of South Atlantic Ocean covering the last glacial (36.4-18.6 ka) to reconstruct temperature and hydroclimate changes, both of which were linked with latitudinal movements of the southern hemisphere westerlies. PCA was used to reduce the dimensionality of multi proxy data. Isotope-enabled GCM was also included to investigate the controlling factors on precipitation d2H in the study area. The new South Atlantic record was compared with other paleoclimate records from northern hemisphere and other regions (both on continents and in oceans) to discuss the interhemispheric links during the last glacial. The manuscript was overall mature and well-written, but but the section 5 of this manuscript is a bit hard to read, particularly

for me to eyeballing Figs. 4, 6, 10, 11 at the same time. The contents and orders of these figures might be further improved.

Specific comments:

L90 & L92: The setting of site is vague. It was a "overgrown crater lake" but also part of a "peat bog"?

L131 & L147: Could you explain what k-value is and how it is related to estimate sedimentation rate?

L264: if BIT is close to 1 steadily throughout the core, it does not mean any prerequisite for the valid use of brGDGTs-based proxies. BIT itself does not endorse anything considering its complexity in lake sediments. It only indicates low concentrations of crenarchaeol in the samples you analyzed. It could be that the paleo-lake is small and shallow without in-situ lake thaumarchaea community, or that the catchment soils are generally wet (Dirghangi et al., 2013).

L277-278: It is indeed possible and not surprising for lake sediments. This is why brGDGT data from adjacent catchment soils are needed to verify how much similarity in brGDGT compositions is between top sediments and adjacent soils, although the situation could be different during the last glacial and between different depositional environments (gyttja, peat, etc.). The brGDGT source is very important and could have shifts over long time scales. The TOC profile you presented suggested the lake sediments are "gyttja" over this period, but the input of soil organic matter seems still likely. If the percentage between lake OM and soil OM shifted over time, the organic geochemistry data might be tricky.

L300: Southern Annular Mode is a more popular term.

L304: Are you just using the raw proxy data as the input to run PCA? Have you standardized proxy data or logarithmized proxy data? Is it going to influence PCA results?

L386-390: It is very unclear to me. The influence of marine animals on organic matter

d15N needs some reference citation. Is the signal from bird guano? Why marine bird signals are suggesting "more or less continuous of SHW"? Are winds driving upwelling and bring nutrients to harbor seabirds? Why rising d13C values are related with more "aquatic" organic carbon (many freshwater plants can have negative d13C)? How can you infer "higher influence from C4 grasses" when you have already said "after which time aquatic sources become more important"? What's the modern catchment vegetation composition?

L395: I agree with that n-alkane d2H data has good correspondence with diatom data. You may also have to think of the differential response between mid- and long-chain alkane d2H. Both precipitation/source water d2H and evaporative enrichment can cause d2H shifts, but it is possible to separate these two signals by using the difference between terrestrial and aquatic plant biomarker d2H (Seki et al., 2011; Rach et al., 2014). Again, it is better to have some modern plant biomarker data at this site.

L414 & 428: You cannot use MST/MAT as a proxy for "winter temperature". Pearson et al. (2011) gave a transfer function for summer air temperature for the reason that many of their lakes have strong seasonality: only summer is biologically important and only summer temperature is monitored. Most of intact brGDGT molecules were produced during summer. Again, winter has almost nothing to do with brGDGT-producing bacteria as the whole lake is frozen and catchment is covered by snow. Loomis et al. (2012) chose mean annual air temperature because most of their lakes are from high-elevation tropics with very weak seasonality: every season is biologically important. Your study site is in subtropical mid-latitude and I expect your brGDGT data will be controlled by mean annual air temperature with a bit bias towards warmer/more productive summer, but you cannot just use that ratio to represent "winter temperature". This is a misunderstanding on GDGT proxy itself.

L441, Fig. 7A: during the late Holocene simulation, the natural variability of d2H has a range of ∼10 permil. Using correspondent maximum zonal wind latitude data, you can estimate the sensitivity of precipitation d2H to westerly core belt latitude, in unit

of permil/degree latitude. This might be useful for paleo-westerly reconstruction. You may also note that the range of sediment core n-alkane d2H data are much larger than ~10 permil, indicating that (1) millennial-scale westerly latitudinal shifts have much larger amplitude than inter-annual scale of the late Holocene, and/or (2) sediment core d2H data, especially "aquatic" n-C23 have been dominated by variations in evaporative enrichment in lake water.

L432: The isotope GCM is simulating precipitation d2H variation, but not the lake water evaporation signal, although it is possible that drier conditions could cause increased raindrop re-evaporation below cloud, just like today's Nevada and Arizona, but I am not sure if raindrop process is included in isotope GCM. The higher d2H values in model might indicate increasing contribution of low latitude moisture. If the westerly moved south, the island will be close to expanding subtropical zone, leading to increasing likelihood influenced by descending and warm (less negative d2H) air masses. If the westerly moved north, the island will be close to westerly core, receiving more moisture from polar air (more negative d2H), more similar to today's 50 degree S condition.

L447: Is period of "1981-2010" within 20th century reanalysis? Should it be 1901-2000? Is the gray lines in (B)-(D) are showing results from 20th century? Figure caption is incomplete.

L612: Why frost can be linked with weathering and CIA index?

L670: Is there any physical mechanism to explain the transition from unstable to stable climate mode?

Technical corrections

L35: 4 k

L162: Xavg, avg subscript

L181: and 15N/14N

[Figure]

L193-194: bracket was missing

---

## Referee Comment (RC2) · Anonymous Referee #2 · 31 Jul 2019

This paper presents a comprehensive, multi-proxy analysis of a sediment core from South Atlantic Nightingale Island. The data are used to reconstruct past hydroclimate, temperature and Southern Hemisphere westerly winds. The authors then explore interterhemispheric linkages, including evidence for DO events and the bipolar see saw connecting Greenland and Antarctic records, and relationships between past SHW strength and atmospheric CO2.

Abstract 23 The abstract is a series of rather unrelated statements. It needs to be re-written following a standard structure, e.g.: 1. What problem did you study and why is it important? 2. What methods did you use to study the problem? 3. What were

your key findings? 4. What did you conclude based on these findings and what are the broader implications?

45 ...(SHW) are a....

48 ...fluxes through physical...

75 This paragraph need to end with a clear statement of the aims of the paper – and how they will be addressed. Aims need to be presented in a logical order. For example using hydroclimate and temperature reconstructions to (1) reconstruct changes in the SHW in the Atlantic sector, (2) Identify interhemispheric linkages including evidence for DO events and the bipolar see saw linking Greenland and Antarctic records, and (3) determining if there is a link between past SHW strength and atmospheric CO2. Followed by a statement of why Nightingale Island is an ideal place to address these questions.

109 For each of the methods sections it would be helpful to state why the analysis was carried out in the leading sentence. E.g. on lines 285 and 291 there is no indication of why these analyses are being carried out.

133 Add something about the treatment of 14C outliers (in grey) on Figure 3. These are all younger ages so require an explanation. Lines 323-337 also avoids addressing this issue.

176 Provide a reference for this procedure.

328 Figure 4. It would be useful to have a common zoning system across all stratigraphic figures. The three PCA zones (line 489) dominate the discussion so I suggest using these here. It is not clear what the solid and dashed vertical black lines are on this figure – please explain in the caption. I strongly recommend plotting the 'productivity' indicators as fluxes (Cyperaceae pollen. Terrestrial diatoms, BSi, TOC) as this should provide a more accurate reconstruction of productivity through time.

399-347 These statements would be better placed in the methods. See comments on

Line 109 (above)

351 This statement needs qualifying. There are very few peak by peak similarities with EDML in these records – however I can see some reflection of the 3 PCA zones (line 489) across the different proxies.

357 State where these step changes are.

368 Figure 5. Please include a cluster analysis on this figure and also superimpose the PCA zones so that readers can see if the PCA zones are reflected in the pollen data. Ditto the diatom data (Fig. S2).

406 PCA – use capitals, cf. 417-418

555 and 622 Include the PCA zones on these figures as these are cited throughout the discussion.

581 State age and depth of this transition

586 Mark Antarctic LGM on figure

591 Replace 'good correspondence' with 'some correspondence'

604-609 The relationships with CO2 merit a separate subheading

630-653 This section could be strengthened by referring back to the original stated aims of the paper (see comments on Line 75 above).

676-678 This interpretation is not well-supported as the main phase of deglaciation was well after 18.6 Ka (see: Bentley, M. J., Ó Cofaigh, C., Anderson, J. B., Conway, H., Davies, B., Graham, A. C., Hillenbrand, C.-D., Hodgson, D. A., Larter, R. D., Mackintosh, A., and Verleyen, E.: A community-based geological reconstruction of Antarctic Ice Sheet deglaciation since the Last Glacial Maximum, Quaternary Science Reviews, 100, 1-9, 2014).

---

## Author Comment (AC1) · 22 Aug 2019

We thank for many useful comments and as you will see in the attached file we will change the text and figures according to the comments.

Please also note the supplement to this comment: https://www.clim-past-discuss.net/cp-2019-65/cp-2019-65-AC1-supplement.zip

---

## Author Response (AR1)

General comments: Björck et al. generated a multi-proxy record from a sediment core in a mid-latitude island of South Atlantic Ocean covering the last glacial (36.4- 18.6 ka) to reconstruct temperature and hydroclimate changes, both of which were linked with latitudinal movements of the southern hemisphere westerlies. PCA was used to reduce the dimensionality of multi proxy data. Isotope-enabled GCM was also included to investigate the controlling factors on precipitation d2H in the study area. The new South Atlantic record was compared with other paleoclimate records from northern hemisphere and other regions (both on continents and in oceans) to discuss the interhemispheric links during the last glacial. The manuscript was overall mature and well-written, but the section 5 of this manuscript is a bit hard to read, particularly for me to eyeballing Figs. 4, 6, 10, 11 at the same time. The contents and orders of these figures might be further improved.

Specific comments: L90 & L92: The setting of site is vague. It was a "overgrown crater lake" but also part of a "peat bog"?

L131 & L147: Could you explain what k-value is and how it is related to estimate sedimentation rate?

L264: if BIT is close to 1 steadily throughout the core, it does not mean any prerequisite for the valid use of brGDGTs-based proxies. BIT itself does not endorse anything considering its complexity in lake sediments. It only indicates low concentrations of crenarchaeol in the samples you analyzed. It could be that the paleo-lake is small and shallow without in-situ lake thaumarchaea community, or that the catchment soils are generally wet (Dirghangi et al., 2013).

L277-278: It is indeed possible and not surprising for lake sediments. This is why brGDGT data from adjacent catchment soils are needed to verify how much similarity in brGDGT compositions is between top sediments and adjacent soils, although the situation could be different during the last glacial and between different depositional environments (gyttja, peat, etc.). The brGDGT source is very important and could have shifts over long time scales. The TOC profile you presented suggested the lake sediments are "gyttja" over this period, but the input of soil organic matter seems still likely. If the percentage between lake OM and soil OM shifted over time, the organic geochemistry data might be tricky.

L300: Southern Annular Mode is a more popular term.

L304: Are you just using the raw proxy data as the input to run PCA? Have you standardized proxy data or logarithmized proxy data? Is it going to influence PCA results?

L386-390: It is very unclear to me. The influence of marine animals on organic matter d15N needs some reference citation. Is the signal from bird guano? Why marine bird signals are suggesting "more or less continuous of SHW"? Are winds driving upwelling and bring nutrients to harbor seabirds? Why rising d13C values are related with more "aquatic"

**Kommentar [s1]:** We understand that it may be difficult to eyeball those figures at the same time, but some also belong to the early result section and therefore come earlier. We realized very early that this may be a problem and have therefore tried our best to place the figs in an optimal and logical order. In addition, a few curves are shown more than once.

**Kommentar [s2]:** The long-term development of a not too deep lake is that it turns into a peat bog through over-growth, especially if there is time enough! We have changed the text.

**Kommentar [s3]:** The k-value is the parameter in the sediment deposition model which determines the number of deposition events per unit depth. In practice, this determines the variability of sedimentation rates and hence, the rigidity of the age-depth model. If k is set to a small value, the modelled sedimentation rates are allowed to be very variable, while bigger values will lead to more constant accumulation rates, and hence, a "more stiff" age-depth model. The k-parameter is well known to all OxCal users and extensively described and discussed in [Bronk Ramsey 2008, Quaternary Science Reviews, 27 (1-2), 42-60]. We added the reference in the text, so that the interested reader who is not aware of the concept can find more information on it.

**Kommentar [s4]:** Agreed. We therefore wrote ONE prerequisite, not THE prerequisite. It is not clear what the reviewer likes us to change here, but we changed to: 'A basic prerequisite for the use of...'

**Kommentar [s5]:** We agree with the reviewer. Given the 'cold bias' in the soil calibration for lakes (=warm bias for soils when using a lake calibration) it is possible that our record reflects changing sources. However, we do not find an obvious correlation between GDGT-derived temperature and two proxies for terrestrial influx, the C/N ratio and magnetic susceptibility. Lake calibration sets will already have a certain soil component in them, with some lakes in that set having more, others less soil-derived GDGTs. In any case, soil and lake-derived calibrations do move in the same general direction even though the slope and intercept are different. The most pragmatic approach is still to use a lake-based calibration set, and see to what extent it corroborates with other proxy evidence. Our reconstruction is in very good agreement with the Antarctic temperature reconstruction and also covaries with the other proxies used … [1]

**Kommentar [s6]:** Changed.

**Kommentar [s7]:** We have used the raw data and variables were centered and standardized, as stated in chapter 3.13.

organic carbon (many freshwater plants can have negative d13C)? How can you infer "higher influence from C4 grasses" when you have already said "after which time aquatic sources become more important"? What's the modern catchment vegetation composition?

L395: I agree with that n-alkane d2H data has good correspondence with diatom data. You may also have to think of the differential response between mid- and longchain alkane d2H. Both precipitation/source water d2H and evaporative enrichment can cause d2H shifts, but it is possible to separate these two signals by using the difference between terrestrial and aquatic plant biomarker d2H (Seki et al., 2011; Rach et al., 2014). Again, it is better to have some modern plant biomarker data at this site.

L414 & 428: You cannot use MST/MAT as a proxy for "winter temperature". Pearson et al. (2011) gave a transfer function for summer air temperature for the reason that many of their lakes have strong seasonality: only summer is biologically important and only summer temperature is monitored. Most of intact brGDGT molecules were produced during summer. Again, winter has almost nothing to do with brGDGT-producing bacteria as the whole lake is frozen and catchment is covered by snow. Loomis et al. (2012) chose mean annual air temperature because most of their lakes are from highelevation tropics with very weak seasonality: every season is biologically important. Your study site is in subtropical mid-latitude and I expect your brGDGT data will be controlled by mean annual air temperature with a bit bias towards warmer/more productive summer, but you cannot just use that ratio to represent "winter temperature". This is a misunderstanding on GDGT proxy itself.

L441, Fig. 7A: during the late Holocene simulation, the natural variability of d2H has a range of ~10 permil. Using correspondent maximum zonal wind latitude data, you can estimate the sensitivity of precipitation d2H to westerly core belt latitude, in unit of permil/degree latitude. This might be useful for paleo-westerly reconstruction. You may also note that the range of sediment core n-alkane d2H data are much larger than ~10 permil, indicating that (1) millennial-scale westerly latitudinal shifts have much larger amplitude than inter-annual scale of the late Holocene, and/or (2) sediment core d2H data, especially "aquatic" n-C23 have been dominated by variations in evaporative enrichment in lake water.

L432: The isotope GCM is simulating precipitation d2H variation, but not the lake water evaporation signal, although it is possible that drier conditions could cause increased raindrop re-evaporation below cloud, just like today's Nevada and Arizona, but I am not sure if raindrop process is included in isotope GCM. The higher d2H values in model might indicate increasing contribution of low latitude moisture. If the westerly moved south, the island will be close to expanding subtropical zone, leading to increasing likelihood influenced by descending and warm (less negative d2H) air masses. If the westerly moved north, the island will be close to westerly core, receiving more moisture from polar air (more negative d2H), more similar to today's 50 degree S condition.

L447: Is period of "1981-2010" within 20th century reanalysis? Should it be 1901- 2000? Is the gray lines in (B)-(D) are showing results from 20th century? Figure caption is incomplete.

L612: Why frost can be linked with weathering and CIA index?

**Kommentar [s8]:** We have now deleted some text and clarified some text and added a reference.

**Kommentar [s9]:** A modern plant dataset would be useful but only a full and complete study would give more insight than what is already known from other studies, and that falls outside the scope of this already very data-rich paper. On top, this is impossible to get anymore. Moreover this will reflect a modern plant community and not that of the LGM or before
Yes we have thought about the differential repsonse between terrestrial and aquatic dD. This can go various ways, as it depends a lot on the lake size, hydrology, climate, and terrestrial ecosystem. The approach in Rach et al and Seki et al is only valid when lakes are relatively insensitive to dryness and higher plants are more sensitive. However lake water (especially small shallow lakes like the one in this study) itself can also evaporate under dry summer conditions and thus change isotopically , whereas terrestrial lipids can be less sensitive because of a good groundwater reservoir that acts as a buffer. This situation turns the sensitivity towards drought around compared to the one used in Rach et al. - and is actually the sa… [2]

**Kommentar [s10]:** We fully agree with the reviewer that this may not be a valid approach as a proxy for winter temperature. We have therefore deleted MST/MAT in the text, figures and P… [3]

**Kommentar [11]:** We think that a linear interpretation of the relationship between isotopic composition of precipitation and latitude is very risk… [4]

**Kommentar [12]:** This is a very good comment and we have changed the text accordingly.

**Kommentar [s13]:** We have noted the importance of evaporative enrichment for n-C23 in the text already.

**Kommentar [14]:** This comment is assigned to wrong line number, possibly to lines below l. 453.

**Kommentar [15]:** Yes, this is included in the model, and now also in the text.

**Kommentar [16]:** This is what we mean on l. 461-465, but we have clarified it.

**Kommentar [17]:** We only use this period due to lack of SH data to constrain the reanalysis prior to the satellite era

**Kommentar [18]:** Yes, model in black and 20CR in gray. Figure caption is now completed.

**Kommentar [s19]:** The CIA index is linked to weathered material and mechanical weathering increases with temp changes causing periods below and above freezing.

L670: Is there any physical mechanism to explain the transition from unstable to stable climate mode? Technical corrections

L35: 4 k

L162: Xavg, avg subscript

L181: and 15N/14N

L193-194: bracket was missing

**Kommentar [s20]:** As we have noted slightly below it may be explained by the fact that we move from MIS3 to MIS2, and e.g. that the Subtropical Front (SF) moved north of Tristan, but the large scale processes behind this global scale change is still under debate.

**Kommentar [s21]:** Shouldn´t there be an "a" to designate years? We think so.

**Kommentar [s22]:** Changed

**Kommentar [s23]:** Changed

**Kommentar [s24]:** Changed

| Sid. 1: [1] Kommentar [s5] | seb | 2019-08-21 16:01:00 |
|---|---|---|

We agree with the reviewer. Given the 'cold bias' in the soil calibration for lakes (=warm bias for soils when using a lake calibration) it is possible that our record reflects changing sources. However, we do not find an obvious correlation between GDGT-derived temperature and two proxies for terrestrial influx, the C/N ratio and magnetic susceptibility. Lake calibration sets will already have a certain soil component in them, with some lakes in that set having more, others less soil-derived GDGTs. In any case, soil and lake-derived calibrations do move in the same general direction even though the slope and intercept are different. The most pragmatic approach is still to use a lake-based calibration set, and see to what extent it corroborates with other proxy evidence. Our reconstruction is in very good agreement with the Antarctic temperature reconstruction and also covaries with the other proxies used in the PCA. In fact, when we now have removed the MAT and MAT/MST from the PCA we can compare the PC1 data with the MAT values and it shows a corr coeff of 75%! This is now included in the text. We therefore argue for a predominant influence of temperature, not source changes. To avoid confusion, we now only use the one from Loomis et al in the main text. We have adjusted the wording in our text reflecting the concerns of the reviewer, in line with the reply given here.

| Sid. 2: [2] Kommentar [s9] | seb | 2019-08-21 16:04:00 |
|---|---|---|

A modern plant dataset would be useful but only a full and complete study would give more insight than what is already known from other studies, and that falls outside the scope of this already very data-rich paper. On top, this is impossible to get anymore. Moreover this will reflect a modern plant community and not that of the LGM or before

Yes we have thought about the differential repsonse between terrestrial and aquatic dD. This can go various ways, as it depends a lot on the lake size, hydrology, climate, and terrestrial ecosystem. The approach in Rach et al and Seki et al is only valid when lakes are relatively insensitive to dryness and higher plants are more sensitive. However lake water (especially small shallow lakes like the one in this study) itself can also evaporate under dry summer conditions and thus change isotopically , whereas terrestrial lipids can be less sensitive because of a good groundwater reservoir that acts as a buffer. This situation turns the sensitivity towards drought around compared to the one used in Rach et al. - and is actually the same as the intepretation from the reviewer on the large dD variability of C23 (see below). We therefore refrain from using the approach by Rach and discuss the (covarying)  terr and aquatic dD values separately and interpret the dD as mainly caused by circulation changes. The reviewer does not appear to be opposed by this so we leave our dD discussion as it is.

| Sid. 2: [3] Kommentar [s10] | seb | 2019-08-21 16:10:00 |
|---|---|---|

We fully agree with the reviewer that this may not be a valid approach as a proxy for winter temperature. We have therefore deleted MST/MAT in the text, figures and PCA. We have also deleted MAT in the PCA, as stated above.

| Sid. 2: [4] Kommentar [11] | Unknown Author | 2019-08-20 10:22:00 |
|---|---|---|

We think that a linear interpretation of the relationship between isotopic composition of precipitation and latitude is very risky due to the very dynamical processes in this region.

Clim. Past Discuss.,
https://doi.org/10.5194/cp-2019-65-RC2, 2019

[Figure]

This paper presents a comprehensive, multi-proxy analysis of a sediment core from South Atlantic Nightingale Island. The data are used to reconstruct past hydroclimate, temperature and Southern Hemisphere westerly winds. The authors then explore interhemispheric linkages, including evidence for DO events and the bipolar see saw connecting Greenland and Antarctic records, and relationships between past SHW strength and atmospheric CO2.

Abstract 23 The abstract is a series of rather unrelated statements. It needs to be rewritten following a standard structure, e.g.: 1. What problem did you study and why is it important? 2. What methods did you use to study the problem? 3. What were

your key findings? 4. What did you conclude based on these findings and what are the broader implications?

. . .(SHW) are a. . ..

. . .fluxes through physical. . .

This paragraph need to end with a clear statement of the aims of the paper – and how they will be addressed. Aims need to be presented in a logical order. For example using hydroclimate and temperature reconstructions to (1) reconstruct changes in the SHW in the Atlantic sector, (2) Identify interhemispheric linkages including evidence for DO events and the bipolar see saw linking Greenland and Antarctic records, and (3) determining if there is a link between past SHW strength and atmospheric CO2. Followed by a statement of why Nightingale Island is an ideal place to address these questions.

For each of the methods sections it would be helpful to state why the analysis was carried out in the leading sentence. E.g. on lines 285 and 291 there is no indication of why these analyses are being carried out.

Add something about the treatment of 14C outliers (in grey) on Figure 3. These are all younger ages so require an explanation. Lines 323-337 also avoids addressing this issue.

Provide a reference for this procedure.

Figure 4. It would be useful to have a common zoning system across all stratigraphic figures. The three PCA zones (line 489) dominate the discussion so I suggest using these here. It is not clear what the solid and dashed vertical black lines are on this figure – please explain in the caption. I strongly recommend plotting the 'productivity' indicators as fluxes (Cyperaceae pollen. Terrestrial diatoms, BSi, TOC) as this should provide a more accurate reconstruction of productivity through time.

399-347 These statements would be better placed in the methods. See comments on

**Kommentar [s1]:** The abstract is now totally altered.

**Kommentar [s2]:** Changed

**Kommentar [s3]:** Changed

**Kommentar [s4]:** We have now added this good suggestion into the text.

**Kommentar [s5]:** This has now been done for most of the methods.

**Kommentar [s6]:** We have now added explanations for the outliers.

**Kommentar [s7]:** This is something that is made routinely for C/N analyzes to account for differences in atomic weight: 14(N)/12(C) = 1.167. Added in the text.

**Kommentar [s8]:** This is done!

**Kommentar [s9]:** We find it unnecessary to explain that it shows every 500 and 1000 years.

**Kommentar [s10]:** We partly understand this point. For some of the proxies (e.g BioSi and TOC) it is impossible to calculate fluxes since we do not have dry density and since productivity is not the focus of the paper this can be done in a separate, more palaeo-ecologically focused paper, where we can present different types of pollen and diatom data. So we keep it as % for now, since we e.g. think that the relative impact of Cyperaceae may have influence on the alkanes.

[Figure]

Line 109 (above)

This statement needs qualifying. There are very few peak by peak similarities with EDML in these records – however I can see some reflection of the 3 PCA zones (line 489) across the different proxies.

State where these step changes are.

Figure 5. Please include a cluster analysis on this figure and also superimpose the PCA zones so that readers can see if the PCA zones are reflected in the pollen data. Ditto the diatom data (Fig. S2).

PCA – use capitals, cf. 417-418

and 622 Include the PCA zones on these figures as these are cited throughout the discussion.

State age and depth of this transition

Mark Antarctic LGM on figure

Replace 'good correspondence' with 'some correspondence'

604-609 The relationships with $CO_2$ merit a separate subheading

630-653 This section could be strengthened by referring back to the original stated aims of the paper (see comments on Line 75 above).

676-678 This interpretation is not well-supported as the main phase of deglaciation was well after 18.6 Ka (see: Bentley, M. J., Ó Cofaigh, C., Anderson, J. B., Conway, H., Davies, B., Graham, A. C., Hillenbrand, C.-D., Hodgson, D. A., Larter, R. D., Mackintosh, A., and Verleyen, E.: A community-based geological reconstruction of Antarctic Ice Sheet deglaciation since the Last Glacial Maximum, Quaternary Science Reviews, 100, 1-9, 2014).

**CPD**

———————

Interactive
comment

[Figure]

**Kommentar [s11]:** This is done and by a more detailed description of the methods and their utility.

**Kommentar [s12]:** Done! We have reformulated the similarities.

**Kommentar [s13]:** Done!

**Kommentar [s14]:** We refer once again to the point that the focus is not paleoecology, so for our purpose we only use selected pollen data for climate reconstructions. Therefore independent zonation of the pollen diagram (cluster analysis) is not relevant. Also, the pollen data does not indicate any major vegetation changes except for a few temp sensitive taxa. For the diatom data we concentrate on the ratios and not the percentages. So we think it is unnecessary, and the PCA zones are very clear in the diatom diagram without cluster analysis.

**Kommentar [s15]:** Done!

**Kommentar [s16]:** Done!

**Kommentar [s17]:** Done!

**Kommentar [s18]:** Antarctic LGM is not that well-defined! So we avoid it but discuss it in the text.

**Kommentar [s19]:** Done!

**Kommentar [s20]:** We disagree since this is only a small part of the discussion.

**Kommentar [s21]:** Good idea, but we moved it to the first part of the last section.

**Kommentar [s22]:** We have reformulated this so it is understood as the initial/start of the deglaciation.

**A SOUTH ATLANTIC ISLAND RECORD UNCOVERS SHIFTS IN**

**WESTERLIES AND HYDROCLIMATE DURING THE LAST GLACIAL**

**Svante Björck**[1,2]**, Jesper Sjolte**[1]**, Karl Ljung**[1]**, Florian Adolphi**[1,3]**, Roger Flower**[4]**, Rienk**
**H. Smittenberg**[2]**, Malin E. Kylander**[2]**, Thomas F. Stocker**[3]**, Sofia Holmgren**[1]**, Hui Jiang**[5]**,**
**Raimund Muscheler**[1]**, Yamoah K. K. Afrifa**[6]**, Jayne E. Rattray**[7]**, Nathalie Van der**
**Putten**[8]

[1]Department of Geology, Lund University, SE-22362 Lund, Sweden
[2]Department of Geological Sciences and the Bolin Centre for Climate Research, Stockholm
University, SE-10691 Stockholm, Sweden
[3]University of Bern, Physics Institute, Climate and Environmental Physics, Sidlerstrasse 5, CH-3012
Bern, Switzerland
[4]Department of Geography, University College London, London WC1E 6BT, UK
[5]Key Laboratory of Geographic Information Science, East China Normal University, 200062
Shanghai, PR China
[6]School of Geography, Earth and Environmental Sciences, University of Birmingham, Edgbaston, B15
2TT, UK
[7]Department of Biological Sciences, University of Calgary, Calgary, Canada
[8]Earth and Climate Cluster, Faculty of Science, Vrije Universiteit, Amsterdam, The Netherlands

**Correspondence:** Svante Björck (svante.bjorck@geol.lu.se)

**Abstract**

Changes in the latitudinal position and strength of the Southern Hemisphere Westerlies (SHW) are thought to be tightly coupled to important climate processes, such as cross- equatorial heat fluxes, Atlantic meridional overturning circulation (AMOC), the bipolar seesaw, Southern Ocean ventilation and atmospheric $CO_2$ levels. However, many uncertainties regarding magnitude, direction, and causes and effects of past SHW shifts still exist due to lack of suitable sites and scarcity of information on SHW dynamics, especially from the Last Glacial. Here we present a detailed hydroclimate multi-proxy record from a

36.4-18.6 ka old lake sediment sequence on Nightingale Island (NI). It is strategically located at 37°S in the central South Atlantic (SA) within the SHW belt and situated just north of the marine Subtropical Front (SF). This has enabled us to assess hydroclimate changes and their link to the regional climate development as well as to large-scale climate events in polar ice cores. The NI record exhibits continuous impact of the SHW, recording shifts in both position and strength, and between 36-31 ka the westerlies show high latitudinal and strength-wise variability possibly linked to the bipolar seesaw. This was followed by 4 ka of slightly falling temperatures, decreasing humidity and fairly southerly westerlies. After 27 ka temperatures decreased 3-4°C, marking the largest hydroclimate change with drier conditions and a variable SHW position. We note that periods with more intense and southerly positioned

SHW seem to be related to periods of increased $CO_2$ outgassing from the ocean, while changes in the cross-equatorial gradient during large northern temperature changes appear as the driving mechanism for the SHW shifts. Together with coeval shifts of the South Pacific westerlies, it shows that most of the Southern Hemisphere experienced simultaneous atmospheric circulation changes during the latter part of the last glacial. Finally we can conclude that multiproxy lake records from oceanic islands have the potential to record atmospheric variability coupled to large-scale climate shifts over vast oceanic areas.

[revised manuscript text omitted]

**3 Material and methods**

A large set of proxy data was analyzed, including chemical (N, XRF elemental concentrations and isotopes ($^{13}$C, $^{15}$N, $^{2}$H or D)), biological (TOC, molecular fossils such as *n*-alkanes, glycerol dialkyl glycerol tetraether lipids (GDGTs), pollen and diatom assemblages, biogenic silica (BSi)), and physical (magnetic susceptibility (MS)) parameters. Some proxies provide information about local changes such as soil conditions/erosion (C/N ratios, $^{13}$C and MS), weathering (major element data), vegetation composition (pollen, *n*-alkane distributions), organic productivity (TOC and BSi), lake conditions and levels (diatoms, BSi, δD values of short-chained *n*-alkanes) and bird impact ([15]N). Others display regional changes in hydroclimate, such as mean annual air (MAAT) and mean summer air temperatures (MST) from the GDGT lipids and the source water of terrestrial and aquatic plants including evaporative conditions (hydrogen isotopes, δD). Observations of the isotopic content of precipitation are very sparse around TdC, and therefore we have investigated the hydroclimate variability with an isotope enabled climate model. In addition, we have performed principal component analysis (PCA) to distinguish the influence of the different proxies on samples (see Methods). Most of our data is found in the Supplementary data file.

[revised manuscript text omitted]
., 2011). Our record could be biased by a changing ratio of soil- and lake-derived GDGTs, where a greater relative contribution of terrestrial-derived GDGTs would result in a warm bias, if a lake calibration is used. However, we do not find a correlation between GDGT-derived temperature and two proxies for terrestrial influx, the C/N

ratio and magnetic susceptibility, but rather the opposite. We  used two lake calibration sets: a) the one of Pearson et al. (2011), based on a global lacustrine data set and using mean summer temperatures (MST), including samples from nearby South Georgia

Island in the S. Atlantic, and b)  a calibration based on a large data set of East African lakes from different altitudes (Loomis et al., 2012), using mean annual temperatures (MAAT), and which is also applicable outside of East Africa (Loomis et al.,

2012). It is impossible to test which of these two proxy records would reflect past conditions more accurately. However, the two reconstructions strongly co-vary, with a difference between reconstructed MST and MAAT of approximately 5°C.

**3.11 Calculation of insolation values**

A long term numerical solution for Earth´s insolation quantities (Laskar et al., 2004) was used for the insolation values, 37-18 ka at 37°S, and calculated with the Analyseries program.

While the austral winter values (W/m2) were based on mean daily June-August insolation (W/m2), the mean austral summer values were based on the mean daily December-

February insolation.

**3.12 Isotope model simulation**

The isotope model analysis is based on a 1200-year simulation using the isotope enabled version of the ECHAM5/MPIOM earth system model (Werner et al., 2016) run with natural and anthropogenic forcings for 800 to 2000 CE (Sjolte et al., 2018). Horizontal resolution of the atmosphere is $3.75^{o}$ x $3.75^{o}$ (T31) with 19 vertical layers, while the ocean has a horizontal resolution of $3^{o}$ x $1.8^{o}$ with 40 vertical layers. The model includes isotope fractionation for all phase changes in the hydrological cycle, including below cloud evaporation. Since both the present day situation and our Nightingale Island record show a continuous impact from the westerlies we deem it valid to use this late Holocene simulation as an analogue for interpreting the variability of the westerlies during the time period of study. The outcome of the simulation is presented in the result section, but further investigation of the model run shows that the multi-decadal variability of δD at TdC is related to the phase of the

Annular Mode, indicating that isotopic variability at TdC is sensitive to large scale

SH climate variability (Fig. S4).

**3.13 Principal component analysis (PCA)**

PCA was performed with 14  of our proxies (Fig. 6B) that we expect to respond to hydroclimate changes, but without the MAAT values in order to test the MAAT values vs.

other climate proxies, using the C2 program (Juggins, 2007). The aim was to display the impact of different combination of proxies on the samples in a biplot (Fig. 6B), as discussed in Section 4.2.

All proxy data waere centered and standardized before calculation.

**4. Results**

**4. 1 An island record of glacial climate in central South Atlantic**

[revised manuscript text omitted]

values between 30-27.2 ka and with falling and rather low Fe fluxes (Fig. 10F).  We also note higher lake evaporation from δD values of the aquatic $n$-$C_{23}$ (Fig. 4F), in concert with rising summer insolation (Fig. 11A). Around 27.5 ka we see a brief response in some of the proxies to the short DO3 event (Fig. 10A), such as the MAT MAAT and PC2 records (Figs. 11D and

F), which and is also noticeable in e.g. the marine and Brazilian monsoon records (Figs. 10B

and E).

The start of Zone 3 at 26.5 ka (727 cm) constitutes the most drastic change in our record (Figs. 6A and 9) but timing varies between proxies (Fig. 4). MAT MAAT, TOC and

[revised manuscript text omitted]

mid-latitudes until at least 18.6 -ka. This might have been a combined effect of declining summer insolation and northward shifting westerlies (Figs. 11A and F), conveying cold air masses, sea ice and ice bergs far north fromof Weber et al.´s (2014) first peak of iceberg- rafted debris from collapsing Antarctic ice shelves starting at 20 ka, and named MWP-19KA.

(Weber et al., 2014).

**Author contributions.** S.B. was the initiator of the study, received funds, drilled and described cores,
carried out sampling and XRF analyses and contributed with most writing, J.S. contributed with
interpreting data, much writing, ran the isotope model experiment (ECHAM5-wiso/MPI-OM) and
analyzed all modeling results, K.L. drilled and described cores, carried out sampling, analyzed C, N,
$^{13}$C, $^{15}$N, pollen and contributed with writing, F.A contributed with the age model and some writing,
R.F. contributed with interpreting and analyzing diatom results and some writing, R.H.S. helped
interpret biomarkers and hydrogen isotopes and contributed with some writing, M.E.K. analyzed XRF
results and contributed with some writing, T.F.S. contributed with creative inputs and some writing,
S.H. sampled and carried out diatom analyzes, H.J. carried out multivariate statistics, Y.K.K.A.
analyzed biomarkers and hydrogen isotopes, R.M. calculated insolation values and contributed with
little writing, J.E.R. carried out biomarker analyses and calibrated the GDGTs and N.V.d.P, carried out
biogenic silica analysis. All commented on the manuscript.

**Acknowledgements.** The co- members of the 2010 Tristan expedition (M. Björck, A. Björk, A.
Cronholm, J. Haile, M. Grignon) and Tristan islanders are gratefully acknowledged for hard work at
sea and on Nightingale I. The isotope enabled climate model, ECHAM5-wiso/MPI-OM, was run at the
AWI Computer and Data Center. We thank M. Werner for helping to set up and run the model
simulations, S. Barker, F. Cruz and C. Millo for providing us with their data, G. Ahlberg for pollen
sample preparations and Å. Wallin for magnetic susceptibility measurements. We are grateful for
financial support, incl. expedition costs, from the Swedish Research Council (VR), the Crafoord
Foundation, the Royal Fysiographic Society, the LUCCI Centre in Lund and the Lund and Stockholm universities. We dedicate this paper to Charles T. Porter, our skipper on his ketch Ocean Tramp, who
challenged all kind of weather in the South Atlantic to retrieve our unique sediment cores. However,
he sadly died suddenly in March 2014 while preparing for our next expedition: a great loss in many
respects but mostly as an invaluable, memorable friend and colleague.

[revised manuscript text omitted]